# Weight Decay Improves Language Model Plasticity

**Tessa Han** [1]   **Sebastian Bordt** [2]   **Hanlin Zhang** [3]   **Sham Kakade** [3]

## Abstract

Large language models are typically trained in two broad phases: pretraining to produce a base model, followed by further training to improve downstream performance. However, hyperparameter optimization and scaling laws are studied primarily from the perspective of the base model's validation loss, overlooking a crucial model property: downstream adaptability. In this work, we study pretraining from the perspective of *model plasticity*, that is, the ability of the base model to successfully adapt to downstream tasks upon additional training. We focus on the role of weight decay, a key regularization parameter during pretraining, and show through systematic experiments that larger weight decay increases the plasticity of the pretrained model, resulting in greater performance gains downstream after fine-tuning. This effect can lead to counterintuitive trade-offs where base models that perform worse after pretraining can perform better after further training. Further investigation of weight decay's mechanistic effects on model behavior reveals that it encourages linearly separable representations, regularizes attention matrices, and reduces overfitting on the training data. Together, these findings highlight the importance of pretrained model plasticity, the limits of using cross-entropy loss as the sole metric for hyperparameter optimization, and the multifaceted role that a single optimization hyperparameter plays in shaping model behavior.

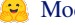 Models      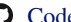 Code

## 1. Introduction

Weight decay is a canonical hyperparameter in deep learning whose role has evolved alongside changes in training

[1]Broad Institute, Schmidt Center [2]University of Tübingen, Tübingen AI Center [3]Harvard University. Correspondence to: Tessa Han <than@broadinstitute.org>, Sham Kakade <sham@seas.harvard.edu>.

*Proceedings of the 43rd International Conference on Machine Learning*, Seoul, South Korea. PMLR 306, 2026. Copyright 2026 by the author(s).

regimes. In classical multi-epoch training, weight decay was understood primarily as a regularizer that improves generalization by shrinking weights and controlling model capacity (Hardt et al., 2016; Zhang et al., 2017; Sun et al., 2025). In contemporary large-scale pretraining, which often involves a single epoch over massive datasets (Brown et al., 2020; Kaplan et al., 2020), weight decay no longer primarily serves the purpose of generalization but plays a decisive role in optimization stability and convergence (D'Angelo et al., 2024; Zhang et al., 2025a; Wang & Aitchison, 2024).

Moreover, modern large language models are typically developed in two distinct stages: a large-scale pretraining phase followed by a post-training phase involving supervised fine-tuning, alignment, and reinforcement learning (Brown et al., 2020; Ouyang et al., 2022; Bi et al., 2024; Lambert et al., 2024). While pretraining and post-training are functionally linked, current practices often treat them as decoupled. Specifically, pretraining hyperparameters and scaling laws are predominantly studied through the lens of the base model's validation loss, under the assumption that a lower pretraining validation loss also yields a more capable downstream model (Hoffmann et al., 2022; Bi et al., 2024). This decoupling is especially pronounced when the two stages are carried out by different teams or at different times: a "best" pretrained model is selected in isolation and only later adapted for downstream use. However, to what extent does optimizing pretraining hyperparameters for pretraining performance also optimize the final, post-trained model's performance?

In this work, we study the relationship between pretraining and post-training from the perspective of *model plasticity*. Model plasticity is the ability of a trained model to effectively adapt to new data upon further training, modifying its parameters and internal representations in response to the new data and enabling effective learning of new tasks without reinitialization (Berariu et al., 2021; Dohare et al., 2024). While the literature on model plasticity and on language models have evolved largely independently, the notion of plasticity naturally bridges pretraining and post-training: while pretraining loss measures how well a model learned the training distribution, plasticity captures how readily that model can be reshaped for downstream tasks. As we show in this work, two models with similar pretraining loss may differ in their plasticity, meaning that optimizing hyperpa-

rameters for pretraining loss alone may not yield the best post-trained model.

In our experiments, we vary the weight decay hyperparameter during pretraining and subsequently evaluate the pretrained model's ability to learn various tasks during fine-tuning. Pretraining is performed for two model families (Llama-2 and OLMo-2), multiple model sizes (up to 4B parameters), and in both the compute-optimal (20 tokens-per-parameter, TPP hereafter) and overtrained (140 TPP) regimes. Fine-tuning is performed across six Chain-of-Thought (CoT) reasoning tasks, five language understanding and commonsense reasoning tasks, and one safety alignment task, and model performance is evaluated using a comprehensive suite of metrics that cover both solution correctness and quality. Our experimental design takes an end-to-end perspective (Qi et al., 2025; Mayilvahanan et al., 2025), aligning pretraining hyperparameter selection with the ultimate objective of maximizing performance after further training. Our contributions are as follows:

- We show that weight decay is a key factor in shaping model plasticity, facilitating adaptation to new tasks during subsequent fine-tuning. In our experiments across a range of model families, sizes, training regimes, downstream tasks, and evaluation metrics, the evidence points toward an optimal pretraining weight decay value larger than the standard default of 0.1. This highlights the potential for re-evaluating standard hyperparameter choices to better account for a model's downstream adaptability.

- We provide one of the first examples showing that optimizing hyperparameters to minimize pretraining validation loss does not necessarily yield the best downstream model performance. Specifically, we show that there is a training regime where larger weight decay values lead to higher pretraining validation loss *and* better downstream performance after fine-tuning.

- We provide a mechanistic perspective on the effect of weight decay on model training dynamics, showing that weight decay encourages linearly separable representations, regularizes attention matrices, and reduces overfitting on the training data. These effects provide a potential explanation for how weight decay preserves the model's ability to flex and learn during subsequent adaptation, thereby sustaining plasticity and improving downstream performance.

Together, these findings highlight the importance of pre-trained model plasticity in the pretrain-then-post-train development pipeline of modern language models, the limits of using cross-entropy loss as the sole metric for hyperparameter optimization during pretraining, and the multifaceted role of a single optimization hyperparameter (weight decay) in shaping model behavior.

## 2. Related Work

Here, we discuss related work on weight decay and model plasticity and how our work contributes new insights.

**Weight decay in language model training.** Weight decay is a standard hyperparameter in language model training and is commonly implemented in conjunction with adaptive optimizers such as AdamW (Loshchilov & Hutter, 2019; Brown et al., 2020; Grattafiori et al., 2024; OLMo Team et al., 2024; Liu et al., 2024). Beyond its classical role in regularization and generalization (Krogh & Hertz, 1991; Zhang et al., 2018; Loshchilov & Hutter, 2019; Zhou et al., 2024), weight decay has also been shown to play other roles in language model training, such as improving optimization and training stability (D'Angelo et al., 2024), shaping the learning rate (Li et al., 2020; Kosson et al., 2024; 2025), controlling the effective step size (Wen et al., 2025), inducing low-rank attention layers (Kobayashi et al., 2024), and increasing forgetting of contaminated benchmark questions (Bordt et al., 2025). Wang & Aitchison (2024) show that the weights of AdamW can be understood as an exponential moving average, and that the weight decay hyperparameter plays a critical role in controlling its time scale. Bergsma et al. (2025) study how to set weight decay to minimize the pretraining loss of language models, finding that lower weight decay improves pretraining loss in the overtrained (high TPP ratio) regime. Kim et al. (2025) show that larger weight decay improves pretraining loss in the multi-epoch setting. In contrast to previous work which primarily focuses on weight decay's effects on the pretrained model, this paper examines how weight decay during pretraining shapes model plasticity.

**Plasticity of deep learning models.** Model plasticity has previously been studied in the contexts of continual learning, transfer learning, and reinforcement learning, settings in which models often undergo multiple rounds of training (Dohare et al., 2024; Klein et al., 2024; Coetzer et al., 2025). Prior works have demonstrated that image models lose plasticity when subjected to additional rounds of training on new data, leading to a decreased ability to learn this new data (Dohare et al., 2024; Lyle et al., 2023; Klein et al., 2024). Various approaches have been developed to improve model plasticity, including shrinking and perturbing model weights at the start of each training round (Ash & Adams, 2020), identifying and re-initializing less-useful model weights during training (Dohare et al., 2024), pushing weights towards initialization weights during training (Kumar et al., 2023), and learning per-connection plasticity strengths among neuron pairs (Miconi et al., 2018). While previous studies have examined how active forgetting and tokenization (Chen et al., 2023; Abagyan et al., 2025) affect

language model plasticity, research on language model plasticity remains underdeveloped. In contrast to these works, this paper investigates the role of weight decay, a standard hyperparameter for language model training, on language model plasticity.

## 3. Background and Methods

In this section, we provide further background, define the research question, and describe the experimental setup.

**Weight decay in AdamW.** Motivated by prior findings that regularization helps vision models maintain plasticity (Dohare et al., 2024), this paper investigates weight decay's role in language model plasticity. We focus on the weight decay hyperparameter $\lambda$ in the AdamW optimizer which, for each optimizer step $t \geq 1$, performs two decoupled updates: a gradient update given by

$$\theta'_t = \theta_t - \gamma_t \hat{m}_t / (\sqrt{\hat{v}_t} + \epsilon) \tag{1}$$

followed by a weight decay update given by

$$\theta_{t+1} = \theta'_t - \gamma_t \lambda \theta_t \tag{2}$$

based on model parameters $\theta$, learning rate $\gamma$, first- and second-order moment estimates of the gradient $\hat{m}$ and $\hat{v}$, and a small constant $\epsilon$ added for numerical stability (Loshchilov & Hutter, 2019). For language model pretraining, the choice $\lambda = 0.1$ has emerged as a kind of default, used in many pretraining runs where the optimization hyperparameters are known (Brown et al., 2020; Touvron et al., 2023; OLMo Team et al., 2024).

**Language model plasticity.** To assess the plasticity of a pretrained model, we fine-tune the model on a task and then measure its performance on this task. The better the performance on this downstream task, the better the pretrained model was able to learn new data during fine-tuning, thus the higher the plasticity of the pretrained model. This approach to measuring model plasticity is consistent with prior literature (Berariu et al., 2021; Dohare et al., 2024).

In this context, we now specify the research question:

> **Research Question.** How does weight decay during language model pretraining affect model plasticity, i.e., the pretrained model's ability to learn new knowledge during subsequent training?

We investigate this research question empirically. We perform experiments that systematically vary weight decay during pretraining, then fine-tune and evaluate the models' performance on various downstream tasks. Our experiments span various model families, model sizes, training regimes

(TPP ratios), fine-tuning tasks, and evaluation metrics. The setup is as follows.

**Pretraining.** We train Llama-2 models on the FineWeb-Edu dataset (Penedo et al., 2024) and OLMo-2 models on the OLMo-Mix-1124 dataset. We vary model size and TPP ratio, training models at the 20 TPP Chinchilla-optimal ratio (Hoffmann et al., 2022) and at the 140 TPP overtrained ratio. This setup yields five model groups: Llama-2-0.5B-20x, Llama-2-1B-20x, Llama-2-4B-20x, OLMo-2-1B-20x, and OLMo-2-1B-140x. For each model group, we pretrain variants with different weight decay.

**Fine-tuning.** We perform supervised fine-tuning (SFT) of the pretrained models on a variety of downstream tasks: CoT reasoning, language understanding and commonsense reasoning, and safety alignment. For CoT reasoning, we perform fine-tuning using six datasets spanning diverse domains: MetaMathQA (math reasoning), MedMCQA (medical reasoning), PubMedQA (biomedical research), MMLUProCoT (various subjects including chemistry, computer science, economics, history, law, etc.), RACE (reading comprehension), and SimpleScaling (math, science, and logical reasoning) (Yu et al., 2023; Pal et al., 2022; Jin et al., 2019; Wang et al., 2024; Lee, 2025; Lai et al., 2017; Muennighoff et al., 2025). For language understanding and commonsense reasoning, we perform fine-tuning using five standard benchmark datasets: HellaSwag, Winogrande, PiQA, Arc-Easy, and Arc-Challenge (Zellers et al., 2019; Sakaguchi et al., 2021; Bisk et al., 2020; Clark et al., 2018). For safety alignment, we perform fine-tuning using the dataset from Bianchi et al. (2024).

**Evaluation of model performance after fine-tuning.** For CoT reasoning tasks, we evaluate the fine-tuned models in a zero-shot manner, prompting them to generate solutions to test set questions, and assess both the correctness and quality of the solutions using six evaluation metrics (Qi et al., 2025).

- **Greedy** (i.e., **Pass@1**): A single deterministic response is generated (temperature = 0). The question is marked correct if this response is correct.

- **Maj@16**, **RM@16**, and **Pass@16**: Sixteen responses are sampled (temperature = 1). The question is marked correct if the majority answer is correct (Maj@16), if the response with the highest outcome reward model (ORM; Skywork-Reward-Llama-3.1-8B-v0.2) score is correct (RM@16), or if any of the responses are correct (Pass@16).

- **Correct Ratio**: Sixteen responses are sampled (temperature = 1). Among questions with at least one correct response, we compute the proportion of correct responses out of the sixteen sampled responses.

- **ORM Score**: In addition to solution correctness, we also

measure solution quality. Sixteen responses are sampled (temperature = 1). Each response is assigned a score using an ORM (Skywork-Reward-Llama-3.1-8B-v0.2) and the average score is computed.

For language understanding and commonsense reasoning tasks, we evaluate models using cloze-style accuracy. For the safety alignment task, we use a harmfulness reward model to evaluate the harmfulness of model outputs (Bianchi et al., 2024).

**Weight decay's effect on model plasticity across hyperparameter settings.** The setup described above investigates weight decay's effect on model plasticity under standard default settings for other hyperparameters. But does the effect of weight decay on model plasticity depend on the choices of other hyperparameters? To investigate this question, we perform sweeps over pretraining and fine-tuning hyperparameters. For pretraining hyperparameters, we pretrain additional OLMo-2-1B-20x models, varying both weight decay and learning rate during pretraining, and then fine-tune these models. For fine-tuning hyperparameters, we use OLMo-2-1B-20x models (which were pretrained with different weight decay values) and vary weight decay, learning rate, and batch size during fine-tuning.

Additional details on the experimental setup are in Appendix B (pretraining), Appendix C (fine-tuning and evaluation), and Appendix E (weight decay's effect on model plasticity across hyperparameter settings).

# 4. Weight decay Improves Language Model Plasticity

We present the main experimental results. We begin by identifying the optimal pretraining weight decay based on pretraining performance (Section 4.1), a common way to select pretraining hyperparameters (Hoffmann et al., 2022). Next, we investigate how weight decay shapes the plasticity of the pretrained model and identify its optimal pretraining value based on downstream performance (Section 4.2). Then, we examine whether a model's pretraining performance is fully predictive of its downstream performance (Section 4.3).

## 4.1. The optimal pretraining weight decay based on pretraining validation loss

We first identify the weight decay value that leads to the lowest cross-entropy validation loss after pretraining. This is the value considered optimal by current approaches in hyperparameter optimization for language model pretraining (Bergsma et al., 2025). We pretrain various models by sweeping over different weight decay values and fixing all other hyperparameters (the values for weight decay and other hyperparameters are listed in Appendix B). The valida-

tion cross-entropy loss of these pretrained models is shown in Figure 1.

Small weight decay values ($< 0.1$) during pretraining have little effect on pretraining loss (Figure 1a). In contrast, moderate-to-large values ([0.1, 3]) can either decrease or increase pretraining loss, depending on the setting (Figures 1a-c), while extremely large values (e.g., 10) can substantially degrade pretraining performance (Figure 1a). The observation that weight decay can lower pretraining performance is consistent with prior work on vision transformers (Zhai et al., 2022; Abnar et al., 2021). At 20 TPP, for both Llama-2 and OLMo-2 models, we find that the optimal weight decay parameter is larger than the default of 0.1. In particular, among the weight decay values examined, the optimal weight decay is 0.5 for Llama-2-0.5B-20x and Llama-2-1B-20x (Figure 1a), 0.6 for OLMo-2-1B-20x (Figure 1b), and 1.0 for Llama-2-4B-20x (Figure 1c). However, this relationship changes as training time increases. At 140 TPP, for the OLMo-2-1B-140x model, the default value of 0.1 outperforms (leads to a lower validation loss than) larger values of 0.3 and 1.0 (Figure 1c). This result that overtrained models have a lower optimal weight decay is consistent with previous analyses on weight decay scaling laws which recommend decreasing the value of the weight decay hyperparameter as training time (TPP) increases to optimize for pretraining validation loss (Bergsma et al., 2025).

## 4.2. The optimal pretraining weight decay based on downstream performance

Next, we investigate how weight decay during pretraining affects model plasticity and downstream model performance. We fine-tune the pretrained models from Section 4.1 (which were trained with different weight decay values) on diverse downstream tasks spannning CoT reasoning, language understanding and commonsense reasoning, and safety alignment. Then, we evaluate the final models' performance on these tasks. The average downstream performance of the models after fine-tuning on CoT reasoning tasks is shown in Figure 2. The downstream performance for other tasks is in Appendix C.

Among models that achieved reasonable pretraining validation losses in Section 4.1 (i.e., models that are suitable candidates for subsequent training), higher weight decay during pretraining confers a higher degree of model plasticity, enabling the pretrained model to learn better during fine-tuning and perform better on the fine-tuning task. The results show that models pretrained with weight decay higher than the default 0.1 value perform better on downstream tasks. This finding is consistent across model families (Llama-2 and OLMo-2), model sizes (up to 4B parameters), training regimes (20 TPP and 140 TPP), fine-tuning tasks (twelve tasks spanning CoT reasoning, language understanding and

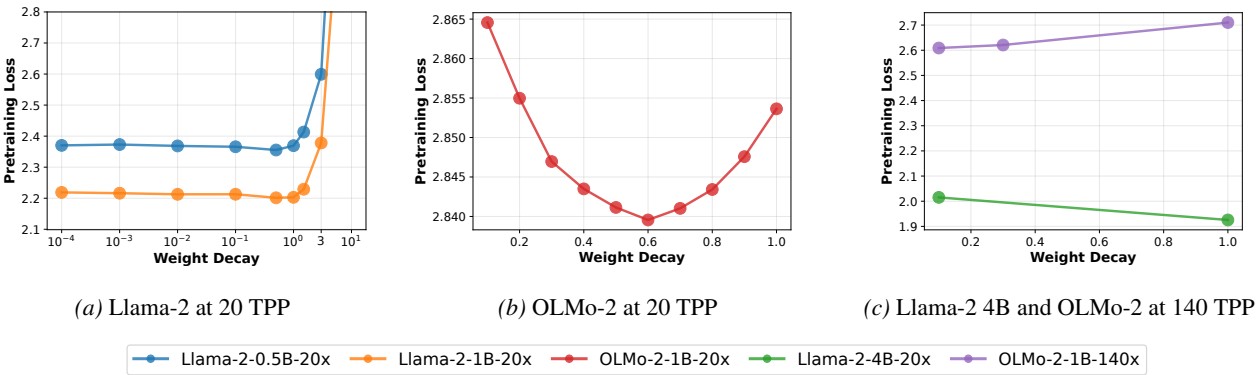

*(a)* Llama-2 at 20 TPP       *(b)* OLMo-2 at 20 TPP       *(c)* Llama-2 4B and OLMo-2 at 140 TPP

*Figure 1.* **Validation cross-entropy loss of models pretrained with different weight decay values.** The weight decay value that minimizes pretraining validation loss may be equal to or larger than the standard default value of 0.1 depending on the training regime.

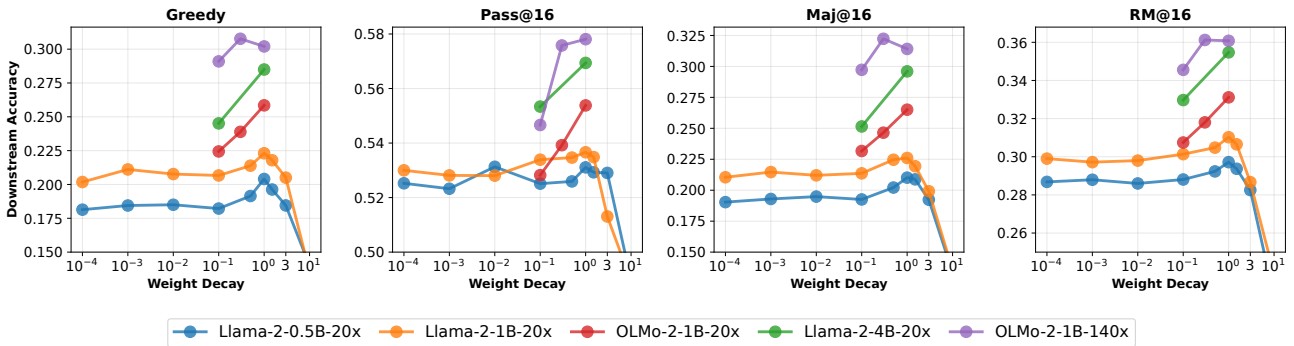

*Figure 2.* **Weight decay during pretraining improves language model plasticity and downstream performance.** We pretrain models with different weight decay values and fine-tune them on various downstream tasks. This figure shows the average downstream performance for six chain-of-thought reasoning tasks. The results indicate that weight decay improves model plasticity and downstream performance. In these experiments, the optimal weight decay for downstream performance is larger than the standard default of 0.1. In addition, the optimal weight decay based on pretraining loss (Figure 1) and that based on downstream performance (this figure) are different, suggesting that optimizing hyperparameters based on pretraining loss alone does not always produce models with the best downstream performance.

commonsense reasoning, and safety alignment), and evaluation metrics. Examining the CoT reasoning tasks more closely, we observe that, among the weight decay values examined, in the compute-optimal 20 TPP regime, the optimal pretraining weight decay is 1.0 (Llama-2-0.5B-20x, Llama-2-1B-20x, Llama-2-4B-20x, and OLMo-2-1B-20x). In the overtrained 140 TPP regime, the optimal pretraining weight decay is 0.3 (OLMo-2-1B-140x). It is possible that, as models are trained for even longer (i.e., beyond 140 TPP), the optimal pretraining weight decay that leads to the best downstream model performance may continue to decrease (this would need to be confirmed with experiments at larger training scales).

We also compare two notions of optimal weight decay: the weight decay value that minimizes pretraining validation cross-entropy loss (Figure 1), as is commonly used in current approaches (Bergsma et al., 2025), and the value that maximizes model performance after fine-tuning (Figure 2). We find that these two weight decay values differ for each

model. We observe that the weight decay value that optimizes model performance after fine-tuning is higher (Llama-2-0.5B-20x, Llama-2-1B-20x, OLMo-2-1B-20x, OLMo-2-1B-140x) or equivalent (Llama-2-4B-20x) to the value that optimizes pretraining validation loss. This shows that the "optimal" weight decay during pretraining is not absolute – it depends on the intended objective, such as optimizing for pretraining performance or downstream performance.

Does weight decay's effect on model plasticity depend on the choices of other hyperparameters? To investigate this, we perform experiments sweeping over pretraining and fine-tuning hyperparameters. We jointly vary weight decay and learning rate during pretraining and weight decay, learning rate, and batch size during fine-tuning. Details and results for these experiments are in Appendix E. Despite varying other hyperparameters, we find that models pretrained with higher weight decay consistently exhibit better downstream performance. Thus, higher weight decay during pretraining consistently improves model plasticity and downstream

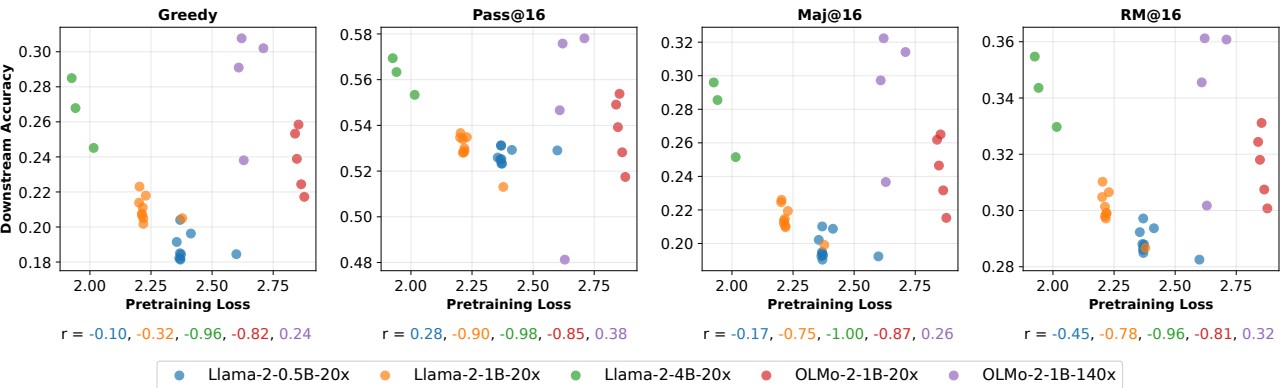

*Figure 3.* **Pretraining performance is not perfectly predictive of downstream performance.** We examine the relationship between the models' pretraining validation cross-entropy loss (x-axis) and downstream accuracy on CoT reasoning tasks (y-axis). Better pretraining performance does not necessarily imply better downstream performance: while the two tend to be correlated, models with similar pretraining losses can perform differently downstream, and models with lower pretraining losses can perform better or worse downstream than models with higher pretraining losses. Thus, optimizing for pretraining loss alone may not always yield the best final model.

performance across a range of pretraining and fine-tuning hyperparameter settings.

> **Finding 1.** Pretraining weight decay can improve model plasticity and lead to better downstream performance. The optimal pretraining weight decay value for plasticity is larger than the default of 0.1.

### 4.3. The relationship between pretraining loss and downstream performance

Following the findings from the previous sections, we now investigate whether a model's pretraining performance is predictive of its downstream performance. We examine the pretraining validation cross-entropy loss of the pretrained models (from Section 4.1) and their downstream accuracy on CoT reasoning tasks after fine-tuning (from Section 4.2). The relationship between these two variables is plotted in Figure 3.

We compare models with the same training setup (i.e., same model family, size, and TPP) that differ only in the pretraining weight decay hyperparameter. Although better pretraining performance tends to be associated with better downstream performance[1], pretraining performance is not a perfect proxy of downstream performance. Sometimes, models with similar pretraining performance can perform differently downstream (such observations exist for Llama-

2-0.5B-20x, Llama-2-1B-20x, and OLMo-2-1B-20x). In addition, models with better pretraining performance (lower pretraining validation loss) can perform better downstream (such observations exist for all five model groups) or worse downstream (such observations exist for Llama-2-0.5B-20x, Llama-2-1B-20x, and OLMo-2-1B-140x). For example, OLMo-2-1B-140x pretrained with weight decay 0.3 or 1.0 performs slightly worse after pretraining (achieving pretraining validation cross-entropy losses of 2.6208 and 2.7064, respectively) than the same model pretrained with weight decay 0.1 (which achieves a pretraining validation cross-entropy loss of 2.6088), but the former two pretrained models perform noticeably better after fine-tuning (Figure 2, purple line). Altogether, these results show that pretraining performance is not perfectly predictive of downstream performance. As a result, optimizing for solely for pretraining loss may not always produce the best downstream (final) models.

> **Finding 2.** The pretraining weight decay value that minimizes the pretraining cross-entropy validation loss does not necessarily lead to the best downstream performance.

## 5. A Mechanistic Perspective on Weight Decay and Model Behavior

Prior work has shown that various factors can influence model plasticity, including the initialization state of model weights at the start of subsequent training, data representation (e.g., tokenization and categorical output representations), and model architecture (e.g., normalization layers) (Ash & Adams, 2020; Abagyan et al., 2025; Lyle et al., 2023). In Section 4, we find that weight decay also shapes

---

[1]The correlations in Figure 3 should be interpreted as suggestive rather than conclusive, given the relatively small sample size of each model group ($n \leq 10$). However, the main takeaway (i.e., better pretraining loss does not necessarily lead to better downstream performance) does not depend on these correlations: in most model groups, there are cases where models with worse pretraining loss achieve better downstream performance, and models with similar pretraining loss perform differently downstream.

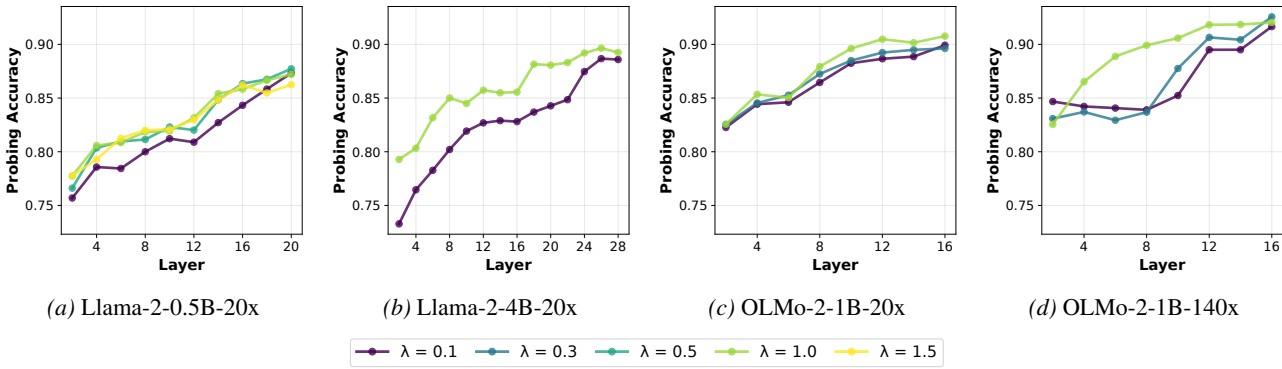

*Figure 4.* **Weight decay encourages linearly separated representations.** This figure depicts the accuracy of linear probes for sentiment and topic for models pretrained with different weight decay values. Stronger weight decay during pretraining results in higher linear probing accuracy, suggesting that weight decay promotes representations that are more linearly separable.

model plasticity. In this section, we explore three mechanisms through which weight decay shapes model behavior: how weight decay shapes the pretrained model's internal representations, attention matrices, and the extent to which it overfits the pretraining data. We also discuss how each mechanism might explain why weight decay improves language model plasticity.

### 5.1. Weight decay encourages linearly separated representations

Inspired by previous findings that weight decay leads to more structured representations in vision models (Jacot et al., 2024), we investigate the effect of weight decay on the representations learned by pretrained language models. We pretrain models with varying weight decay, obtain the last-token embeddings for different types of text at a given model layer, and train a linear probe to classify these embeddings. We examine two tasks: classifying text based on sentiment (positive or negative movie reviews from the Stanford Sentiment Treebank dataset; Socher et al. (2013)) or topic (four types of news articles from the AG News dataset; Zhang et al. (2015)). The average accuracy of these linear probes over the two tasks is shown in Figure 4. Accuracy for individual tasks are in Appendix F.1.

We observe that when a given model is pretrained with higher weight decay, the accuracy of the linear probe trained on the model's representations tends to be higher at every layer of the model. While this relationship is not perfectly monotonic (in some instances, a slightly higher weight decay can lead to a similar or slightly lower probing accuracy), it is generally consistent across weight decay values and model layers. In addition, we observe this relationship across model families, sizes, and training regimes (i.e., for all five model groups). Thus, through these linear probing experiments, we find that representations from models pretrained with higher weight decay result in higher probing

accuracies, indicating that these representations are more linearly separated and suggesting that models pretrained with higher weight decay form more structured internal representations.

The finding that weight decay shapes the representations of pretrained language models points to a potential explanation for why weight decay improves model plasticity (Section 4.2). Pretraining models with higher weight decay produces models with more structured representations, i.e., representations in which information is encoded in a more linearly accessible form. As a result, fine-tuning may focus on refining and aligning existing representations to the fine-tuning task rather than continuing to learn representations, effectively starting at a better initialization and leading to improved downstream performance. This hypothesis is consistent with previous findings that weight decay produces representations that are more transferable to downstream tasks in computer vision (Lee et al., 2023). It is further supported by the observation that the linear separability of model representations (probing accuracy) is strongly positively correlated with downstream model performance (Appendix Figure 25).

### 5.2. Weight decay reduces the rank of attention matrices

Previous work by Kobayashi et al. (2024) provides a theoretical argument that weight decay should reduce the rank of attention matrices. Recall that attention scores can be understood as a bilinear form $X^T W_{QK} X$ where $W_{QK} = W_K^T W_Q \in \mathbb{R}^{n_{embed} \times n_{embed}}$ is the product of the query and key matrices, and $X \in \mathbb{R}^{n_{embed} \times T}$ is the matrix of token embeddings (or hidden representations) for a sequence of length $T$. Now, the matrix $W_{QK}$ is naturally low-rank since its rank is at most $d_{head}$, which is usually significantly smaller than $n_{embed}$. Kobayashi et al. (2024) argue that weight decay should further reduce the rank of $W_{QK}$, as well as of the value-projection matrix

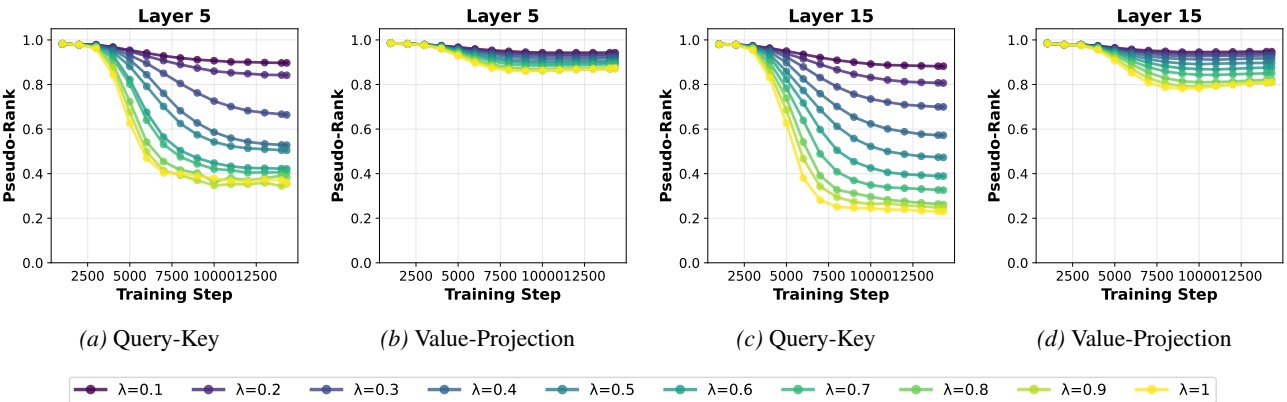

*Figure 5.* **Weight decay reduces the rank of attention matrices.** This figure depicts the average pseudo-rank (Appendix F.2.1) of the query-key ($W_{QK}$) and value projection ($W_{VP}$) matrices in layers 5 and 15 during the training of OLMo-2-1B models at 20 TPP. Stronger weight decay during pretraining leads to lower attention matrix ranks, suggesting that weight decay promotes a more compressed, lower-dimensional attention structure.

$W_{VP} = W_P W_V \in \mathbb{R}^{n_{embed} \times n_{embed}}$. [2] Concretely, they show that L2 regularization applied to the factored matrices $W_K$ and $W_Q$ becomes equivalent to nuclear norm regularization on their product $W_{QK}$, which is known to induce low rank by promoting sparsity in the singular values. While Kobayashi et al. (2024) also provide empirical evidence on the Pile, their experiments were relatively small-scale from today's perspective. We now revisit the impact of weight decay on the rank of attention in our more modern setup.

**Weight decay reduces the rank of attention, but default weight decay yields near full-rank matrices.** Figure 5 depicts the evolution of the pseudo-rank (Appendix F.2.1) of the attention matrices during the training of the OLMo-2-1B-20x models. From Figure 5, we observe that there is a monotonic relationship between the weight decay parameter and the rank of the attention matrices, where larger weight decay values reduce the rank of both $W_{QK}$ and $W_{VP}$. However, unlike what is observed in Kobayashi et al. (2024), we see that the default weight decay parameter of 0.1 yields near full-rank matrices. This observation is further confirmed by Figure 28, which shows that the attention matrices in the fully trained OLMo-2-1B model are nearly full-rank.

**Attention matrices are differently affected by weight decay.** Another important observation from our experiments is that the rank of the matrix $W_{QK}$ seems to be significantly more sensitive to weight decay than $W_{VP}$. In our experiments, a weight decay of 1.0 reduces the rank of $W_{QK}$ by roughly a factor of 2, which is a common rank reduction observed in the literature on low-rank matrices. In contrast, the matrix $W_{VP}$ is still close to full-rank even for a large weight decay value of 1.0. These results are especially pronounced

for Llama-2 models depicted in Figure 26, where the rank of $W_{VP}$ remains essentially stable up to a weight decay value of 1.0, after which the rank collapses—a transition that correlates with a significant drop in performance.

**Low-rank structure as a driver of adaptability.** The observation that increased weight decay leads to lower-rank attention matrices provides a potential explanation for why weight decay improves model plasticity. In machine learning literature, low-rank constraints are a canonical form of regularization that is often believed to encourage simpler, more robust hypotheses (Cai et al., 2010; Oymak et al., 2019; Hu et al., 2022). We conjecture that by encouraging $W_{QK}$ toward a lower-rank configuration, weight decay may prevent the model from overfitting to high-dimensional noise in the pretraining distribution. Such a model may capture higher-level patterns that are more broadly applicable than details specific to the pretraining distribution, thereby enhancing the model's ability to learn new data upon further training and improving downstream performance.

### 5.3. Weight decay reduces overfitting on training data

Lastly, we explore how weight decay influences the extent to which the pretrained model overfits the pretraining data. Previous work has shown that weight decay can cause the forgetting of individual benchmark questions seen during pretraining (Bordt et al., 2025). In the context of model plasticity, the ability to learn new information tends to be associated with the forgetting of prior data, a trade-off commonly referred to as the stability-plasticity dilemma (Kirkpatrick et al., 2017; Riemer et al., 2018; Ibrahim et al., 2024; Elsayed & Mahmood, 2024). Building on these insights, we investigate how weight decay influences overfitting, which is closely related to the forgetting of training data, in pretrained models.

---

[2] $W_{VP}$ is sometimes also denoted as $W_{VO}$ (Wang et al., 2025a;b).

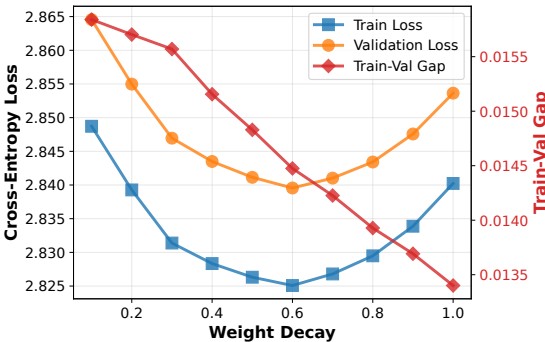

*Figure 6.* **Weight decay reduces overfitting on training data.** This figure depicts the training loss, validation loss, and train-val gap (Equation 3) for the OLMo-2-1B-20x models. As weight decay increases, the train-val gap decreases.

To measure the degree to which a pretrained model overfits the training data, we compute the difference between the loss on the validation data and that on the training data:

$$\text{Train-Val Gap} = \text{Validation Loss} - \text{Training Loss} \quad (3)$$

Here, the training loss is the average loss that the fully trained model encounters on the training data, which is distinct from the training loss curve or the final training loss value. A model that does not overfit the training data would theoretically have a train-val gap of zero. In practice, a larger train-val gap indicates a higher degree of overfitting on the training data, thus less forgetting of the training data.

Figure 6 depicts the train-val gap for the OLMo-2 models trained at 20 TPP. We observe that the train-val gap decreases monotonically as the weight decay parameter is increased. This provides empirical evidence that models trained with larger weight decay values indeed overfit the training data less.

> **Finding 3.** The pretraining weight decay hyperparameter has diverse mechanistic effects on model behavior. It encourages linearly separated representations, regularizes attention matrices, and reduces overfitting on the training data.

## 6. Discussion and Concluding Remarks

This work provides a multidimensional characterization of the effects of the weight decay hyperparameter within the modern language-model training lifecycle. While traditional perspectives have primarily viewed weight decay through the lenses of capacity control in over-parameterized regimes or optimization stability in single-epoch pretraining (Zhang et al., 2017; D'Angelo et al., 2024), our findings suggest that weight decay plays a far more nuanced role in shaping model behavior. In particular, we showed that models with

smaller weight decay achieve lower validation loss after pretraining (especially in the over-trained regime), but that models with larger weight decay benefit from improved plasticity, enabling them to perform best when fine-tuned on downstream tasks. Weight decay may shape model plasticity through several mechanisms, including promoting linearly separable representations, regularizing attention matrices, and reducing overfitting on the training data. Together, these findings reveal fundamental trade-offs in hyperparameter optimization. They also provide one of the first rigorous empirical demonstrations that selecting pretraining hyperparameters based solely on minimal pretraining validation loss can fail to yield the model with the highest performance on downstream tasks.

The trade-offs we show mean that, in practice, the benefits of increased plasticity must be weighed against other effects that may depend on model size, training duration, and other parameters of the training setup. In heavily over-trained scenarios or for very large models trained for many steps (Singh et al., 2025; Comanici et al., 2025; Anthropic, 2025), the benefits of markedly lower pretraining validation loss may outweigh those of plasticity. In addition, weight decay's diverse roles in training dynamics – from plasticity (shown in this work) to optimization, training stability, convergence rate, and overfitting (Hoffmann et al., 2022; D'Angelo et al., 2024; Kosson et al., 2025) – adds further complexity to model training decisions. Its optimal value for one objective can conflict with that for another, as observed when optimizing for pretraining versus downstream performance. A single weight decay value may not satisfy multiple objectives, requiring weighing trade-offs and prioritizing objectives.

Future work may investigate in more detail the trade-offs between stability and plasticity, and the extent to which our results hold in large-model and heavily overtrained scenarios. They may also investigate the role of weight decay in model plasticity for foundation models beyond language (e.g., vision and multimodal foundation models) and for other downstream desiderata (beyond CoT reasoning, language understanding and commonsense reasoning, and safety alignment). Taken together, the findings in this work cast light on the importance of model plasticity, the complexity of hyperparameter tuning throughout the training process of modern language models, and the multifaceted role that a single optimization hyperparameter plays in shaping model behavior.

## Impact Statement

This paper presents work whose goal is to advance the field of machine learning. There are many potential societal consequences of our work, none which we feel must be specifically highlighted here.

## Acknowledgements

We would like to thank Zhenting Qi, Kaiyue Wen, and anonymous reviewers for helpful feedback. SB acknowledges the support from the German Research Foundation through the Cluster of Excellence "Machine Learning – New Perspectives for Science" (EXC 2064/1 number 390727645). SK acknowledges the support from the National Science Foundation Grant under award IIS 2229881. HZ and SK acknowledge the Chan Zuckerberg Initiative Foundation for establishing the Kempner Institute for the Study of Natural and Artificial Intelligence.

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

# Appendix

## Table of Contents

## A. Additional discussion

**Related work.** While this work and Kobayashi et al. (2024) both examine weight decay's effects on attention matrix rank, this work not only replicates previous results in a larger and more modern setting, but also uncovers novel phenomena. Different from Kobayashi et al. (2024), we find that the rank reduction of attention matrices is associated with better pretraining and fine-tuning performance (up to a weight decay of 0.6 for the OLMo models). In addition, we find that the Query-Key and Value-Projection matrices are differently affected by weight decay, which also has not been reported before.

Prior work has also examined how weight decay affects model behavior. For example, Zhang et al. (2025b) studies its role in OOD compositional generalization and reasoning-to-memorization transitions under complexity control. Lee et al. (2023) finds that, in self-supervised learning (SSL), weight decay improves the transferability of image representations, but that the optimal value can be difficult to identify because this benefit is not captured by standard SSL evaluation methods. While our work focuses on language model plasticity, the findings from our work—that higher weight decay can improve downstream performance even when it worsens pretraining loss—and prior works are conceptually consistent.

**Limitations.** We discuss limitations in more detail below.

*Mechanistic explanations are correlational.* The findings that weight decay promotes more linearly separable representations, reduces attention matrix rank, and reduces overfitting (Section 5) serve as potential explanations for why weight decay increases plasticity. However, these mechanisms are correlational in nature. Establishing causality here is challenging because it is hard to disentangle mechanisms from other covariates (e.g., changing attention matrix rank while holding model representations constant). Note that, while the mechanisms are correlational, the effect itself—that weight decay improves plasticity (Section 4)—is causal (since the experiments vary only weight decay and keeps other variables constant).

*Results only apply to the scope examined in the paper.* Our experiments span various model families (Llama-2, OLMo-2), sizes (up to 4B), TPP ratios (20x and 140x), and downstream tasks (six chain-of-thought reasoning tasks, five language understanding and commonsense reasoning tasks, and one safety alignment task). Beyond this scope (e.g., larger models, higher TPP ratios, other fine-tuning tasks, post-training beyond fine-tuning), results may differ.

*Small sample size for some model groups.* We pretrain fewer models for some model groups (Llama-2-4B-20x and OLMo-2-1B-140x) due to the large amount of compute required for their pretraining. However, for these models, the results are consistent with other more extensively studied models: 1) better pretraining validation loss does not guarantee better downstream performance and 2) the weight decay that leads to best downstream performance is larger than the standard 0.1 default.

## B. Pre-training

### B.1. Model architectures and training regimes

We pretrain models from different families (Llama-2 and OLMo-2), of different sizes (up to 4B), and under different training regimes (20 TPP and 140 TPP), yielding the following five model setups: Llama-2-0.5B-20x, Llama-2-1B-20x, Llama-2-4B-20x, OLMo-2-1B-20x, OLMo-2-1B-140x. Model details are in Table 1. For each model setup, we pretrain variants with varying weight decay values.

|  | Llama-2-0.5B | Llama-2-1B | Llama-2-4B | OLMo-2-1B |
|---|---|---|---|---|
| Model size | 0.5B | 1B | 4B | 1.5B |
| Hidden size | 1536 | 2048 | 4096 | 2048 |
| Intermediate size | 3216 | 4896 | 7792 | 16384 |
| Vocab size | 32000 | 32000 | 32000 | 100278 |
| Context length | 2048 | 2048 | 2048 | 4096 |
| # Heads | 32 | 32 | 32 | 16 |
| # Layers | 20 | 22 | 28 | 16 |
| # Query groups | 4 | 4 | 4 | 16 |

*Table 1.* **Model architectures.** We use Llama-2 model architectures from Qi et al. (2025) and OLMo-2 model architecture from OLMo Team et al. (2024). Llama-2 models are trained at 20 TPP and OLMo-2 models are trained at 20 TPP and 140 TPP.

### B.2. Training details

The training data size (measured in tokens) for each model is determined by the TPP ratio.

| Model | Model Size | TPP Ratio | Training Data Size |
|---|---|---|---|
| Llama-2-0.5B-20x | 0.5B | 20 | 10 BT |
| Llama-2-1B-20x | 1B | 20 | 20 BT |
| Llama-2-4B-20x | 4B | 20 | 80 BT |
| OLMo-2-1B-20x | 1.5B | 20 | 30 BT |
| OLMo-2-1B-140x | 1.5B | 140 | 210 BT |

*Table 2.* **Model configurations and training data sizes.**

To pre-train Llama-2 models, we use up to 8 A100 GPUs or 16 H100 GPUs. To pre-train OLMo-2 models, we use 8xH100 GPUs. The OLMo-2-1B-20x models are each trained for 2 days on a single H100 node. The OLMo-2-1B-140x models are trained for 2 weeks on a single H100 node. For all models, we use the AdamW optimizer and standard warmup-cosine learning rate schedule. The only exception is the OLMo-2-1B-140x models, which follow a warmup-stable-decay schedule (Hägele et al., 2024). Llama-2 models are pretrained using the repository from Qi et al. (2025). OLMo-2-1B models are

pretrained using the official repository from AllenAI.

For each model, we train variants with various weight decay values specified in Table 3. Additional training hyperparameters are in Tables 4 and 5.

| Model | Weight Decay |
|---|---|
| Llama-2-0.5B-20x | 10 values: {0, 0.0001, 0.001, 0.01, 0.1, 0.5, 1.0, 1.5, 3.0, 10.0} |
| Llama-2-1B-20x | 10 values: {0, 0.0001, 0.001, 0.01, 0.1, 0.5, 1.0, 1.5, 3.0, 10.0} |
| Llama-2-4B-20x | 3 values: {0, 0.1, 1.0} |
| OLMo-2-1B-20x | 11 values: {0, 0.1, 0.2, 0.3, 0.4, 0.5, 0.6, 0.7, 0.8, 0.9, 1.0} |
| OLMo-2-1B-140x | 4 values: {0, 0.1, 0.3, 1.0} |

*Table 3.* **Weight decay values for each model.** We use the Llama-2-4B-20x weight decay 0.1 model from Qi et al. (2025) and the OLMo-2-1B-140x weight decay 0.1 model from Bordt & Pawelczyk (2025). We pretrain all other models.

| Hyperparameter | Llama-2-0.5B-20x | Llama-2-1B-20x | Llama-2-4B-20x |
|---|---|---|---|
| precision | bf16-mixed | bf16-mixed | bf16-mixed |
| global_batch_size | 512 | 512 | 1024 |
| max_seq_length | 2048 | 2048 | 2048 |
| lr_warmup_ratio | 0.1 | 0.1 | 0.1 |
| max_norm | 1 | 1 | 1 |
| lr | 0.00025 | 0.0002 | 0.00015 |
| min_lr | 0.000025 | 0.00002 | 0.000015 |
| weight_decay | varies | varies | varies |
| beta1 | 0.9 | 0.9 | 0.9 |
| beta2 | 0.95 | 0.95 | 0.95 |
| epoch | 1 | 1 | 1 |

*Table 4.* **Hyperparameters for Llama-2 model pretraining.** For Llama-2 models, hyperparameter values follow those in Qi et al. (2025), except for weight decay, which is varied as the independent variable in our experiments.

| Hyperparameter | OLMo-2-1B-20x | OLMo-2-1B-140x |
|---|---|---|
| precision | bf16-mixed | bf16-mixed |
| global_batch_size | 512 | 512 |
| max_seq_length | 4096 | 4096 |
| lr_warmup_ratio | 0.1 | 0.1 |
| max_norm | 1 | 1 |
| lr | 0.0004 | 0.0004 |
| min_lr | 0.00004 | 0 |
| weight_decay | varies | varies |
| beta1 | 0.9 | 0.9 |
| beta2 | 0.95 | 0.95 |
| epoch | 1 | 1 |

*Table 5.* **Hyperparameters for OLMo-2 model pretraining.** For OLMo-2 models, hyperparameter values follow the OLMo-2 defaults (OLMo Team et al., 2024), except for weight decay, which is varied as the independent variable in our experiments.

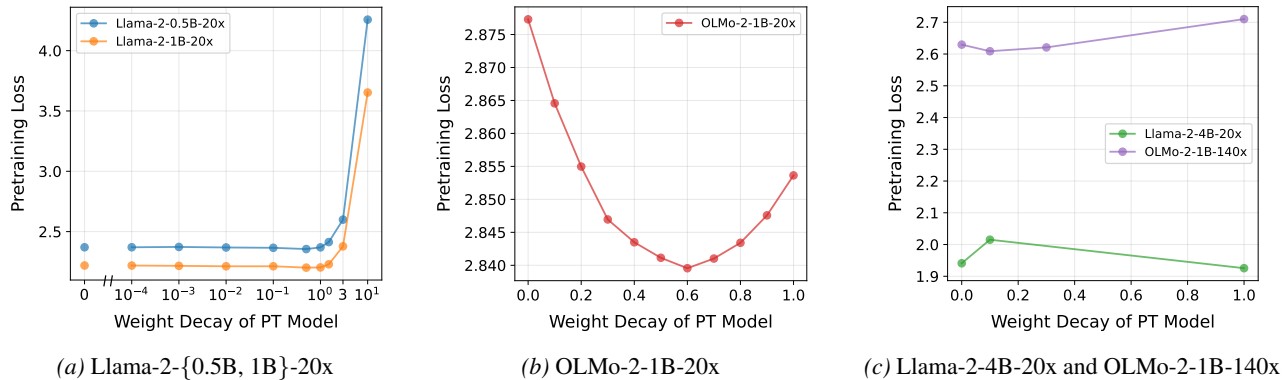

*Figure 7.* **Pretraining validation cross-entropy loss of models pretrained with varying weight decay.** The weight decay value that minimizes pretraining loss may be equal to or larger than the standard default value of 0.1 depending on the training regime.

## C. Fine-tuning

### C.1. Training details

We fine-tune the pretrained models on various downstream tasks (chain-of-thought reasoning, language understanding and commonsense reasoning, and safety alignment) using the following hyperparameters. We use five pretrained models for OLMo-2-1B-20x (weight decay: {0, 0.1, 0.3, 0.6, 1.0}, a range of values including the smallest and largest values, the 0.1 standard default value, and the 0.6 value that led to the lowest pretraining validation loss) and all pretrained models for the other model setups (described in Appendix B.2).

| Hyperparameters | 1B and under | 4B |
|---|---|---|
| cutoff_len | 2048 | 2048 |
| batch_size | 64 | 64 |
| learning_rate | 0.00001 | 0.0000075 |
| lr_scheduler_type | cosine | cosine |
| warmup_ratio | 0.1 | 0.1 |
| n_epochs | 3 | 3 |

*Table 6.* **Hyperparameters for supervised fine-tuning.** We set hyperparameters for fine-tuning following Qi et al. (2025). We set n_epochs = 3 based on results from Qi et al. (2025) showing that this setting leads to the best downstream performance. We use a smaller batch size (batch_size = 64) than Qi et al. (2025) due to computational constraints.

We use the following template for supervised fine-tuning.

*Human: {question}*
*Assistant: {response}*

### C.2. Chain-of-thought reasoning

We fine-tune the pretrained models on six chain-of-thought reasoning datasets. We clean the training data, removing questions that are incoherent or that exceed the models' maximum input sequence length. Information for each dataset is shown in Table 7. Fine-tuned models are evaluated based on answer correctness and quality using the six metrics in Section 3.

| Dataset | Training set | Test set |
|---------|--------------|----------|
| MetaMathQA | $n = 395,000$ | GSM8KPlatinum ($n = 1,209$) + MATH ($n = 5,000$) |
| MedMCQA | $n = 182,555$ | MedMCQA ($n = 4183$) |
| PubMedQA | $n = 211,168$ | PubMedQA ($n = 1000$) |
| MMLUProCoT | $n = 123,836$ | MMLUProCoT ($n = 567$) |
| RACE | $n = 92,737$ | RACE ($n = 4934$) |
| SimpleScaling | $n = 54,484$ | GSM8KPlatinum ($n = 1,209$) + MATH ($n = 5,000$) |

*Table 7.* **Chain-of-thought datasets used for fine-tuning.** MetaMathQA and SimpleScaling are evaluated on test sets of the GSM8KPlatinum (Cobbe et al., 2021; Vendrow et al., 2025) and MATH (Hendrycks et al., 2021) datasets because MetaMathQA and SimpleScaling contain questions that are augmented from the training sets of GSM8KPlatinum and MATH.

Results from experiments are in the figures below.

- Figure 8. Individual and average performance after fine-tuning the pretrained models on chain-of-thought tasks.

- Figure 9. Relationship between pretraining performance (validation cross-entropy loss after pretraining) and performance after fine-tuning for chain-of-thought tasks.

- Figure 10. Stability analysis for observed correlation between pretraining performance and fine-tuning performance.

### C.3. Language understanding and commonsense reasoning

We fine-tune the pretrained models on five language understanding and commonsense reasoning datasets. Information for each dataset is shown in Table 8. Fine-tuned models are evaluated using cloze-style accuracy. Results from experiments are in Figure 11.

| Dataset | Training set | Test set |
|---------|--------------|----------|
| HellaSwag | $n = 39,905$ | $n = 10,042$ |
| Winogrande | $n = 40,398$ | $n = 1,267$ |
| PiQA | $n = 16,113$ | $n = 1,838$ |
| Arc-Easy | $n = 2,251$ | $n = 2,376$ |
| Arc-Challenge | $n = 1,119$ | $n = 1,172$ |

*Table 8.* **Language and general knowledge datasets used for fine-tuning.**

### C.4. Safety alignment

We also examine fine-tuning for safety alignment. Specifically, we fine-tune models on 20,000 general-purpose instructions, randomly sampled from the Alpaca dataset, combined with 300, 500, 1,000, or 2,000 safety-related instructions. Then, we evaluate the fine-tuned models on 100 harmful prompts from the I-MaliciousInstructions dataset. For each generated response, we measure harmfulness using a harmfulness reward model (HRM), which assigns a score from 0 to 4, with higher scores indicating more harmful responses. The experimental setup, training data, test set, and HRM are from Bianchi et al. (2024). Results from experiments for each model are in Figures 12, 13, 14, 15, 16.

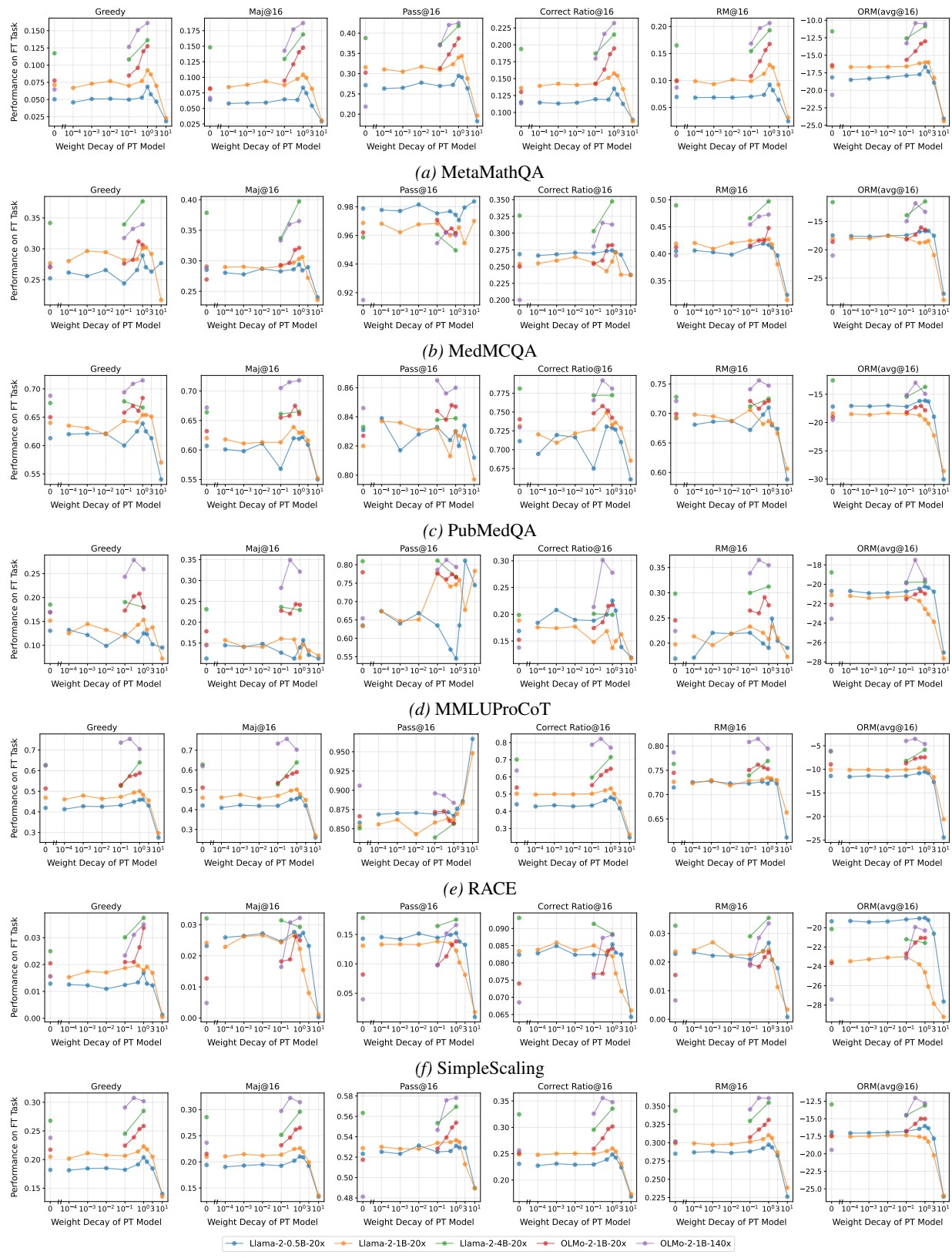

*(a)* MetaMathQA

*(b)* MedMCQA

*(c)* PubMedQA

*(d)* MMLUProCoT

*(e)* RACE

*(f)* SimpleScaling

*(g)* Average over datasets

*Figure 8.* **Fine-tuning performance on chain-of-thought tasks.** Weight decay during pretraining improves model plasticity, leading to higher downstream performance.

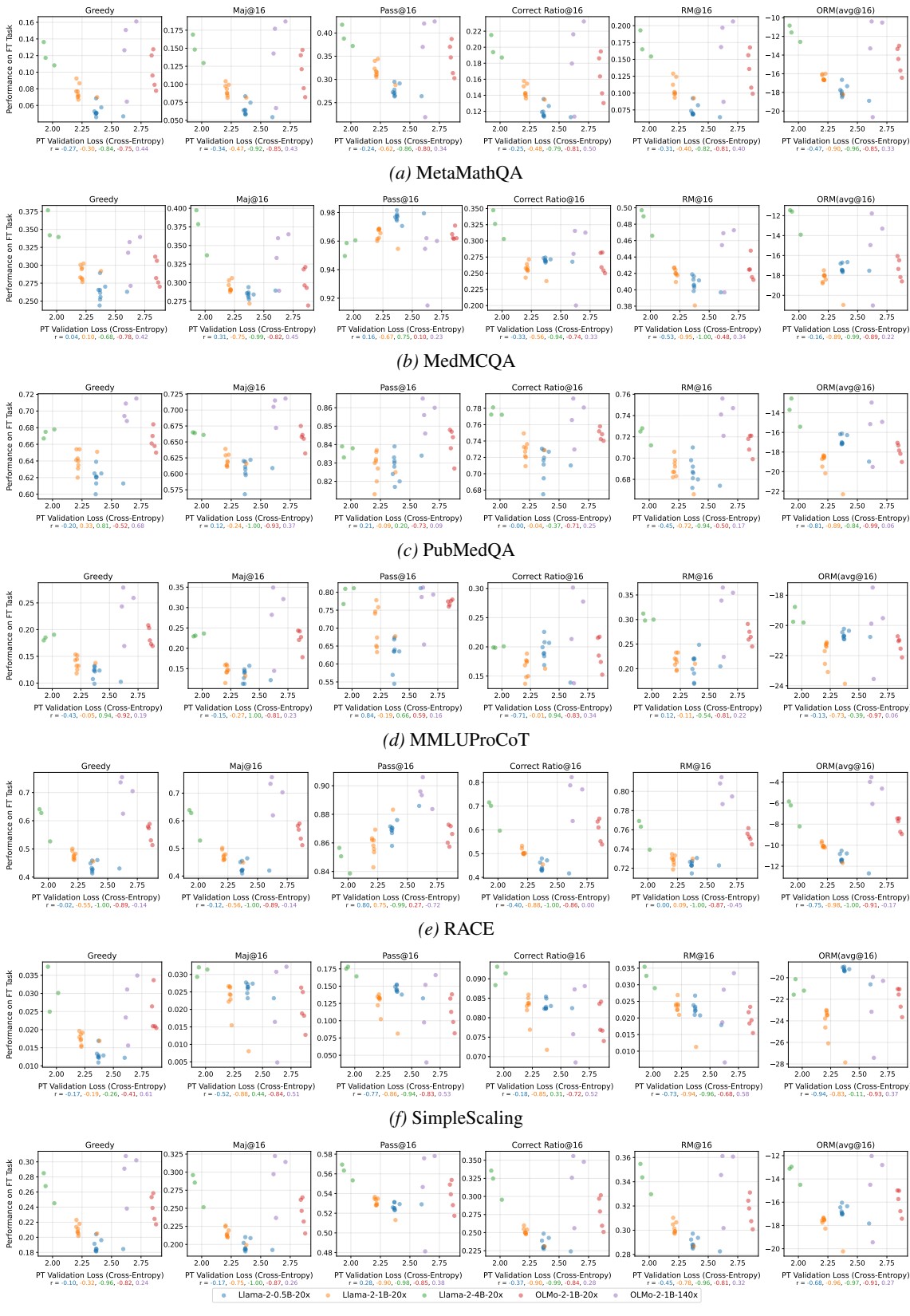

*(a)* MetaMathQA

*(b)* MedMCQA

*(c)* PubMedQA

*(d)* MMLUProCoT

*(e)* RACE

*(f)* SimpleScaling

*(g)* Average over datasets

*Figure 9.* **Pretraining validation cross-entropy loss vs. fine-tuning accuracy for chain-of-thought tasks.** Pretraining validation cross-entropy loss (pretraining performance) is not fully predictive of model performance after fine-tuning (downstream performance).

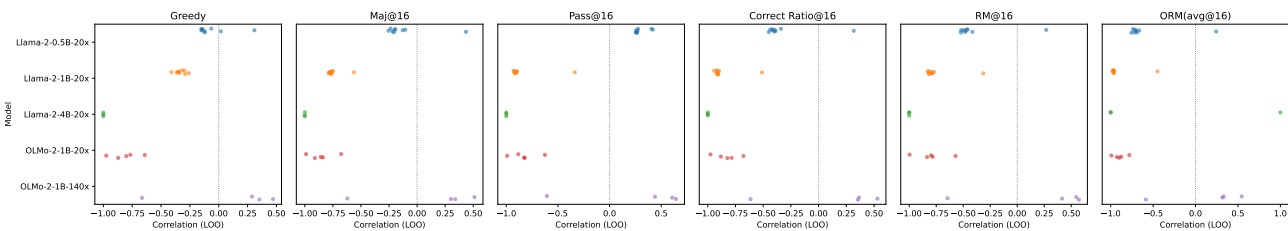

*Figure 10.* **Stability analysis for Pearson correlation coefficient.** Pearson correlation is computed for each leave-one-out (LOO) subset in Figure 9g. The LOO correlation can change noticeably in magnitude and sign, suggesting that the correlation for the full set of data points in Figure 9g is rather unstable, which further supports the finding in that pretraining validation cross-entropy loss (pretraining performance) is not perfectly predictive of fine-tuning accuracy (downstream performance).

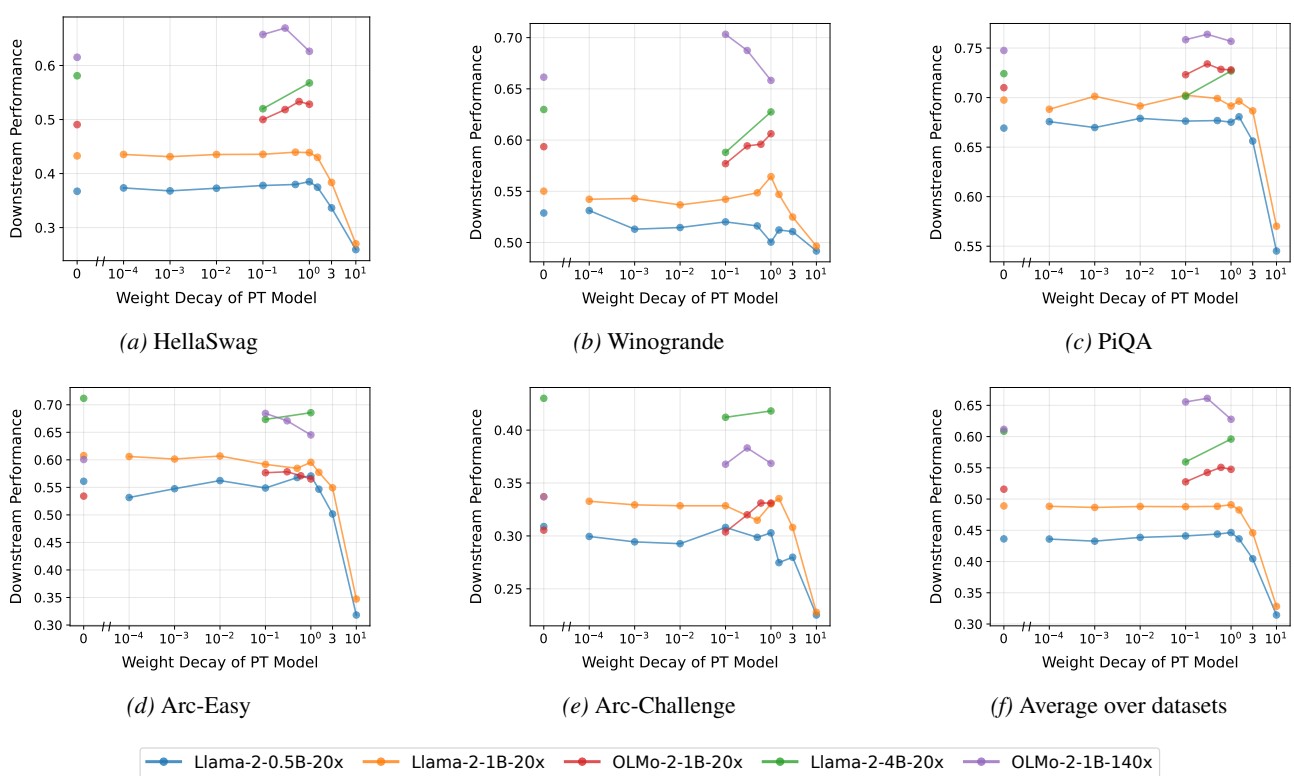

*Figure 11.* **Fine-tuning performance on language and general knowledge tasks.** Weight decay during pretraining improves model plasticity, leading to higher downstream performance.

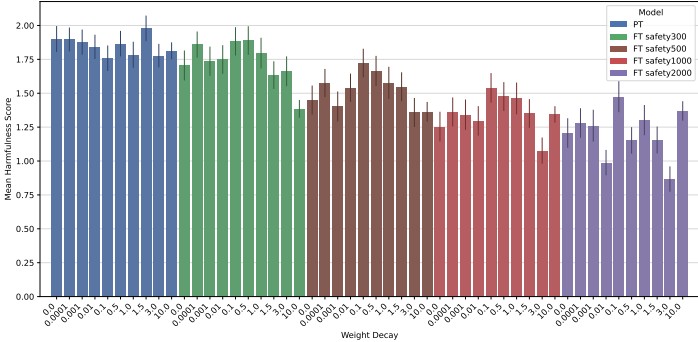

*Figure 12.* **Fine-tuning performance for safety alignment.** Llama-2-0.5B-20x model.

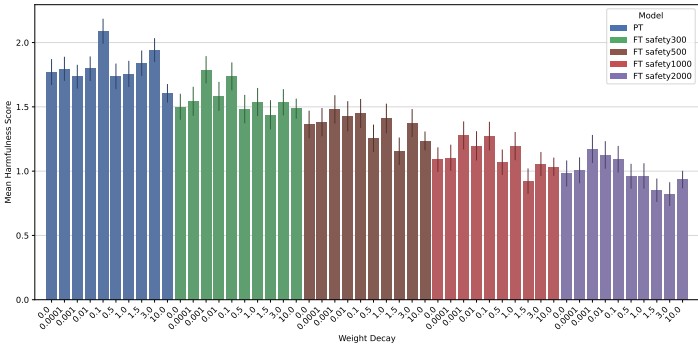

*Figure 13.* **Fine-tuning performance for safety alignment.** Llama-2-1B-20x model.

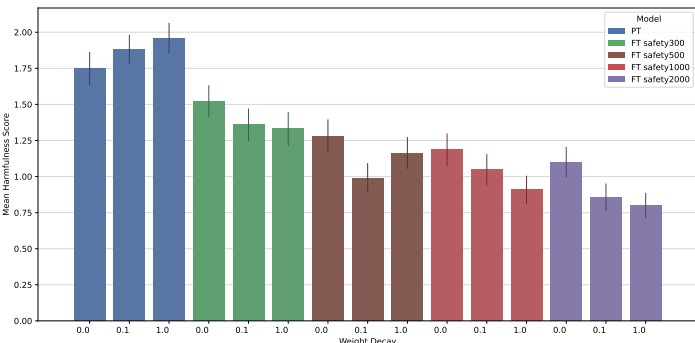

*Figure 14.* **Fine-tuning performance for safety alignment.** Llama-2-4B-20x model.

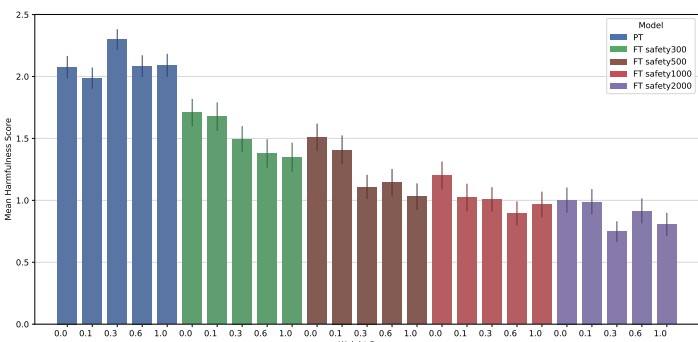

*Figure 15.* **Fine-tuning performance for safety alignment.** OLMo-2-1B-20x model.

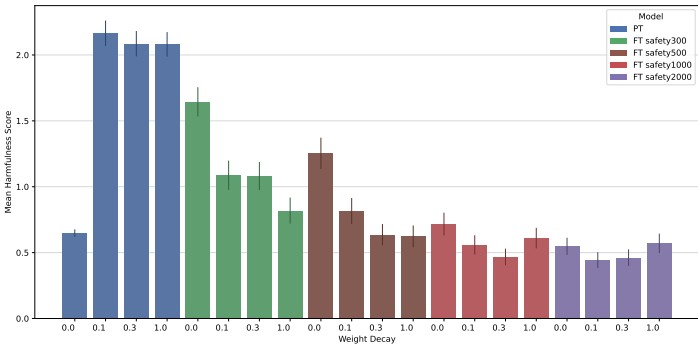

*Figure 16.* **Fine-tuning performance for safety alignment.** OLMo-2-1B-140x model.

# D. Trade-off analyses

**Stability-plasticity trade-off**

As the model adapts to new data during fine-tuning (plasticity), how well does it retain its previous knowledge (stability)? To study this question, we evaluate the upstream performance of the pretrained models and fine-tuned models using five language understanding and commonsense reasoning benchmarks (HellaSwag, PiQA, Winogrande, Arc-Easy, and Arc-Challenge) and measure upstream performance using the average accuracy. We measure stability using the difference in upstream performance between the fine-tuned and pretrained models for a given weight decay value (smaller difference, higher stability) and plasticity using the average downstream performance on the six chain-of-thought reasoning tasks (higher downstream performance, higher plasticity).

We find that while models with higher plasticity also tend to have higher upstream performance after fine-tuning (Figure 17), this higher plasticity sometimes comes with a trade-off in stability (Figure 18).

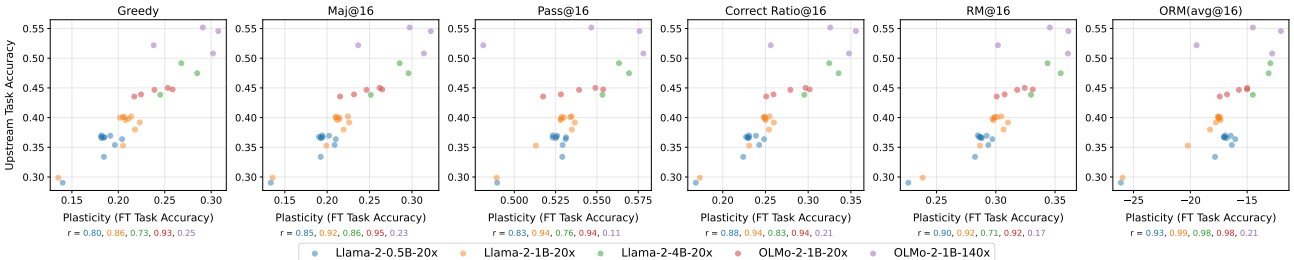

*Figure 17.* **Plasticity vs. upstream accuracy of fine-tuned models.**

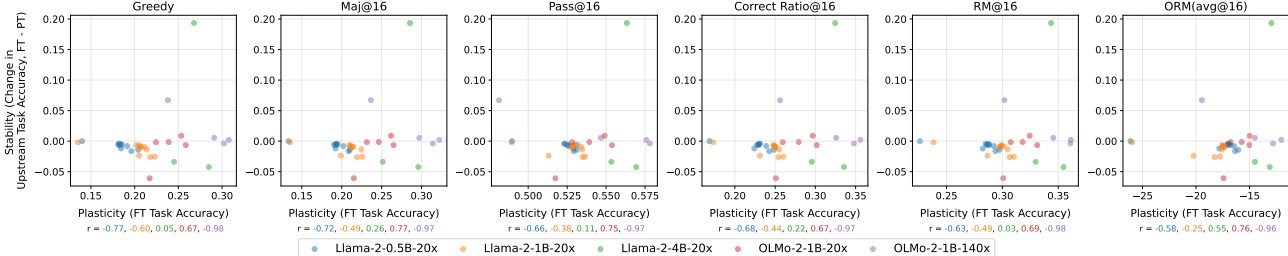

*Figure 18.* **Plasticity vs. stability.**

**Can weight decay hurt upstream performance while improving downstream performance?**

To study this question, we evaluate the upstream performance of the pretrained models using the five language understanding and commonsense reasoning benchmarks. The average upstream accuracies for each model are as follows.

Llama-2-0.5B-20x, weight decay = {0.1, 0.5, 1.0}, upstream accuracy = {0.40, 0.41, 0.41*}

Llama-2-1B-20x, weight decay = {0.1, 0.5, 1.0}, upstream accuracy = {0.44, 0.45, 0.45*}

OLMo-2-1B-20x, weight decay = {0.1, 0.3, 1.0}, upstream accuracy = {0.49, 0.49, 0.50*}

OLMo-2-1B-140x, weight decay = {0.1, 0.3*, 1.0}, upstream accuracy = {0.61, 0.60*, 0.56}

*weight decay value of pretrained model with best downstream performance

We find that while weight decay can lead to more plastic pretrained models that have higher performance on downstream tasks, it does not seem to affect the upstream performance of the pretrained models.

# E. Weight decay's effect on model plasticity across hyperparameter settings

## E.1. Varying pretraining hyperparameters

**Experiment setup.** We examine the effect of pretraining weight decay on model plasticity under varying pretraining hyperparameters. To do so, we pretrain OLMo-2-1B-20x models jointly varying weight decay (wd_pt = [0.1, 0.6, 1.0]) and learning rate (lr = [2e-4, 4e-4, 8e-4]), producing 9 pretrained models. Then, we fine-tune each pretrained model on the six chain-of-thought reasoning datasets, yielding 54 fine-tuned models, and evaluate the performance of the fine-tuned models.

**Results.** We perform the following analyses.

- Figure 19. Effect of pretraining weight decay on model plasticity for a fixed learning rate. We find that, for each learning rate, higher weight decay leads to higher fine-tuning performance. Thus, weight decay's role in model plasticity holds across these pretraining hyperparameter changes. These results support those in Section 4.2 and Appendix C.2, C.3, and C.4.

- Figure 20. Relationship between pretraining and downstream performance. We find that the model with the best pretraining performance is not always the model with the best downstream performance. These results support those in Section 4.3.

## E.2. Varying fine-tuning hyperparameters

**Experiment setup.** We examine the effect of pretraining weight decay on model plasticity under varying fine-tuning hyperparameters. To do so, we fine-tune OLMo-2-1B-20x models pretrained with different weight decay values (wd_pt = [0.1, 0.3, 0.6, 1.0] with default learning rate 4e-4) on two datasets (MetaMathQA and SimpleScaling). During fine-tuning, we jointly vary the learning rate (lr = [1e-5, 3e-5, 6e-5]), weight decay (wd_ft = [0, 0.1, 1.0]), and batch size (bs = [32, 64, 128]). This yields 216 fine-tuned models. Then, we evaluate the performance of the fine-tuned models.

**Results.** We perform the following analyses to examine the effect of pretraining weight decay on model plasticity.

- Figure 21 (MetaMathQA) and Figure 22 (SimpleScaling). Effect of pretraining weight decay for a fixed set of fine-tuning hyperparameters (i.e., for each hyperparameter combination).

- Figure 23 (Rows 1, 3, 5). Effect of pretraining weight decay averaged over hyperparameter combinations.

- Figure 23 (Rows 2, 4, 6). Effect of pretraining weight decay for the best set of fine-tuning hyperparameters (i.e., the fine-tuning hyperparameters that lead to the best fine-tuning performance for a given pretrained model; what one would be most interested in in practice).

In all these analyses, we find that higher pretraining weight decay leads to better fine-tuning performance. Thus, weight decay's role in model plasticity is robust to these changes in fine-tuning hyperparameters.

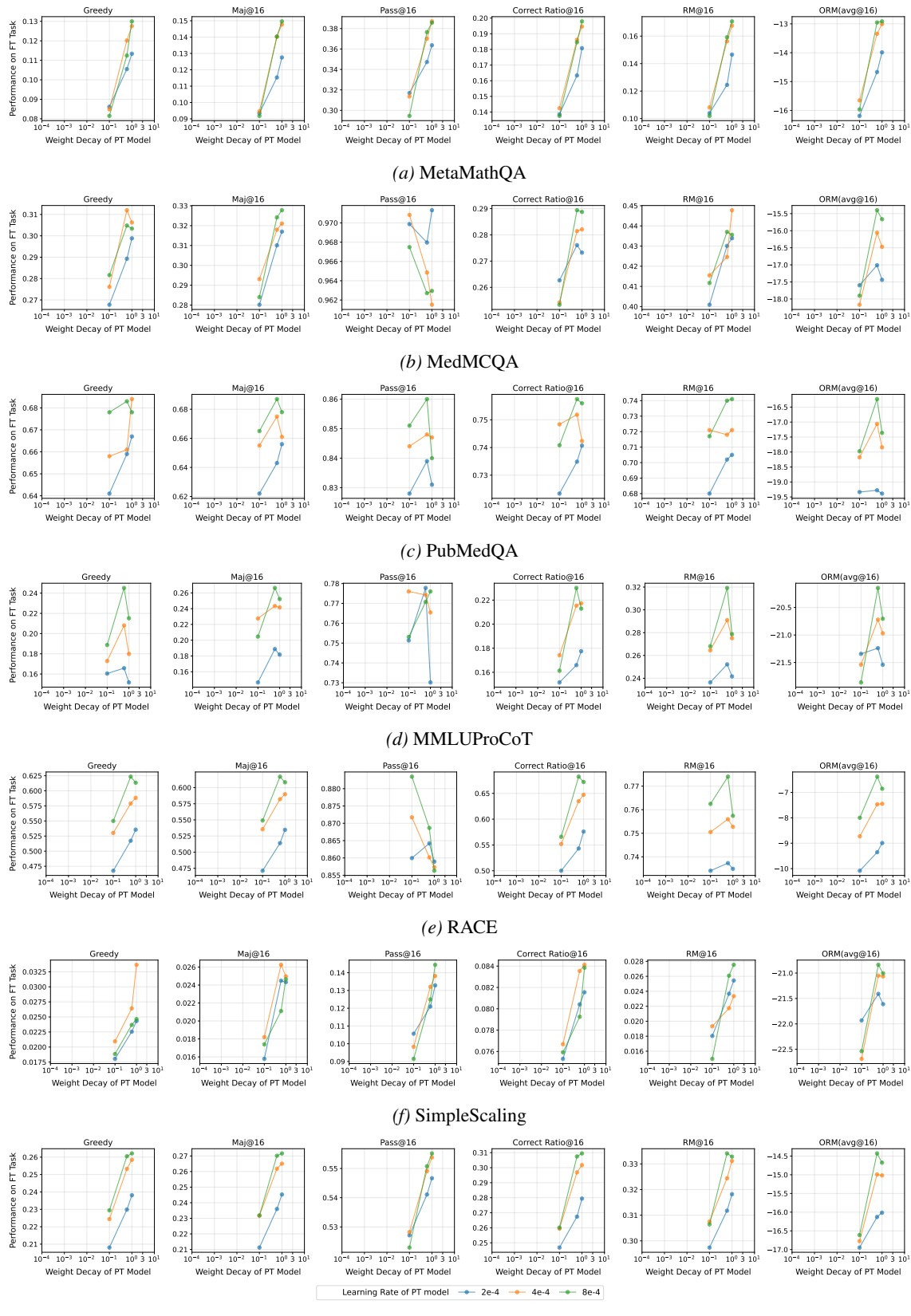

*Figure 19.* **Effect of pretraining hyperparameters on model plasticity.** Models are pretrained with varying weight decay and learning rates. Across learning rates, higher weight decay during pretraining leads to greater model plasticty and higher downstream performance.

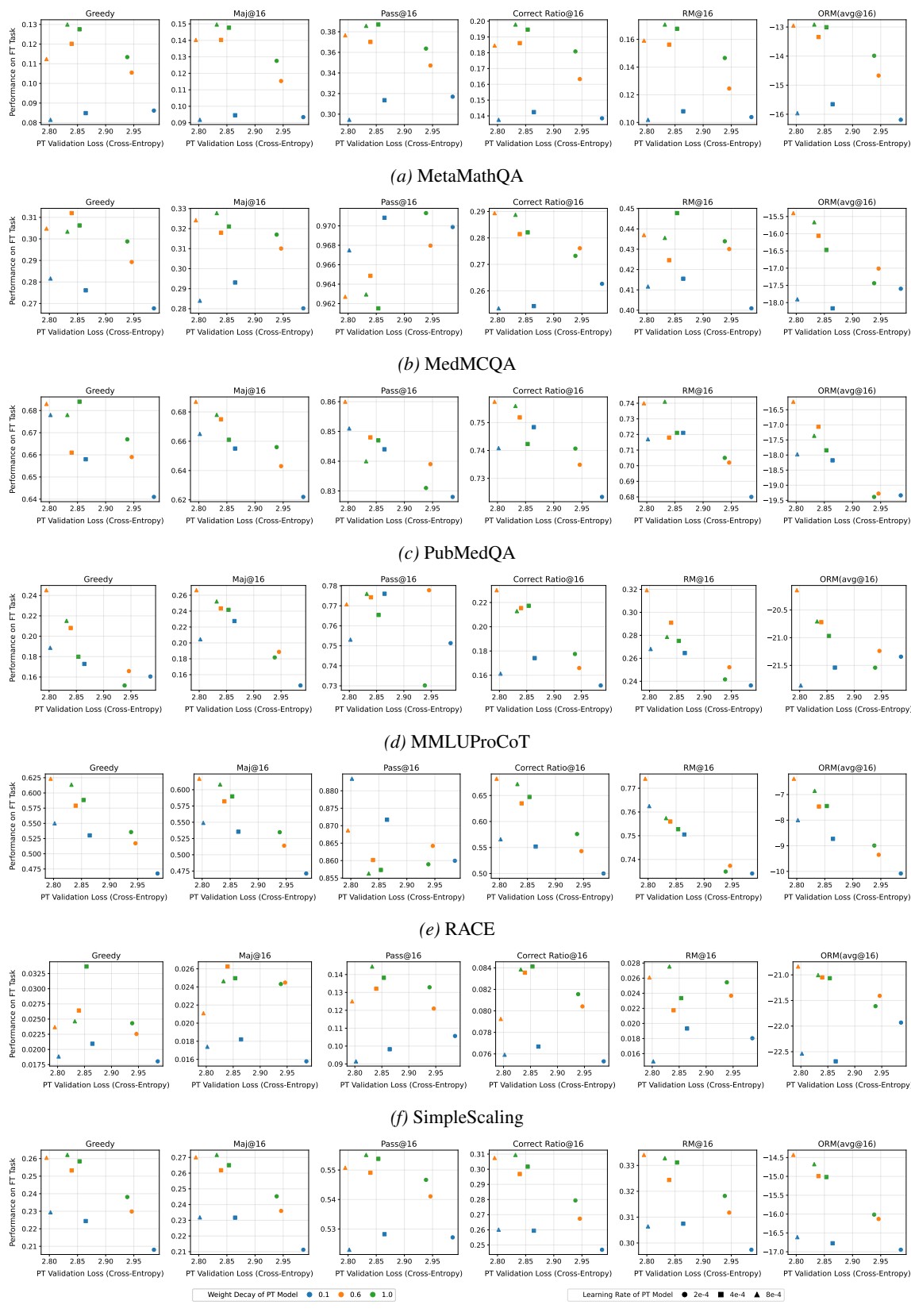

*Figure 20.* **Pretraining (PT) performance is not perfectly predictive of fine-tuning (FT) performance.** Fixing learning rate, the weight decay that leads to the best PT loss does not always lead to the best FT loss. Overall, the model with best PT loss is not the model with best FT loss.

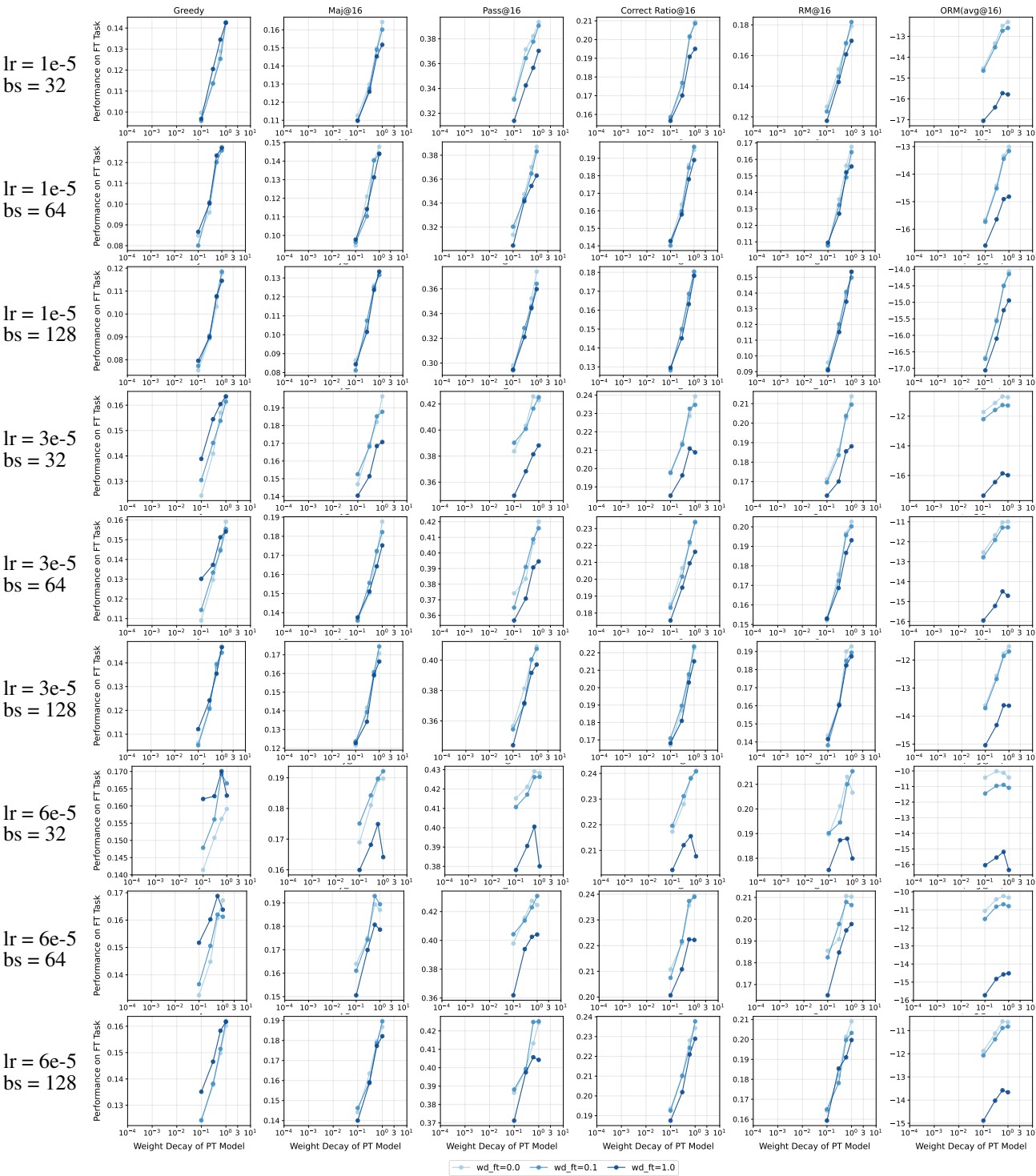

*Figure 21.* **Effect of pretraining weight decay on fine-tuning performance for each set of fine-tuning hyperparameters: Meta-MathQA.** OLMo-2-1B-20x models pretrained with different weight decay values (wd_pt = [0.1, 0.3, 0.6, 1.0]) are fine-tuned on the MetaMathQA dataset. We vary the learning rate (lr = [1e-5, 3e-5, 6e-5]), weight decay (wd_ft = [0, 0.1, 1.0]), and batch size (bs = [32, 64, 128]) during fine-tuning. Average performance over all hyperparameters is shown in Figure 23. Across fine-tuning hyperparameters and evaluation metrics, the higher the weight decay during pretraining, the better the downstream performance.

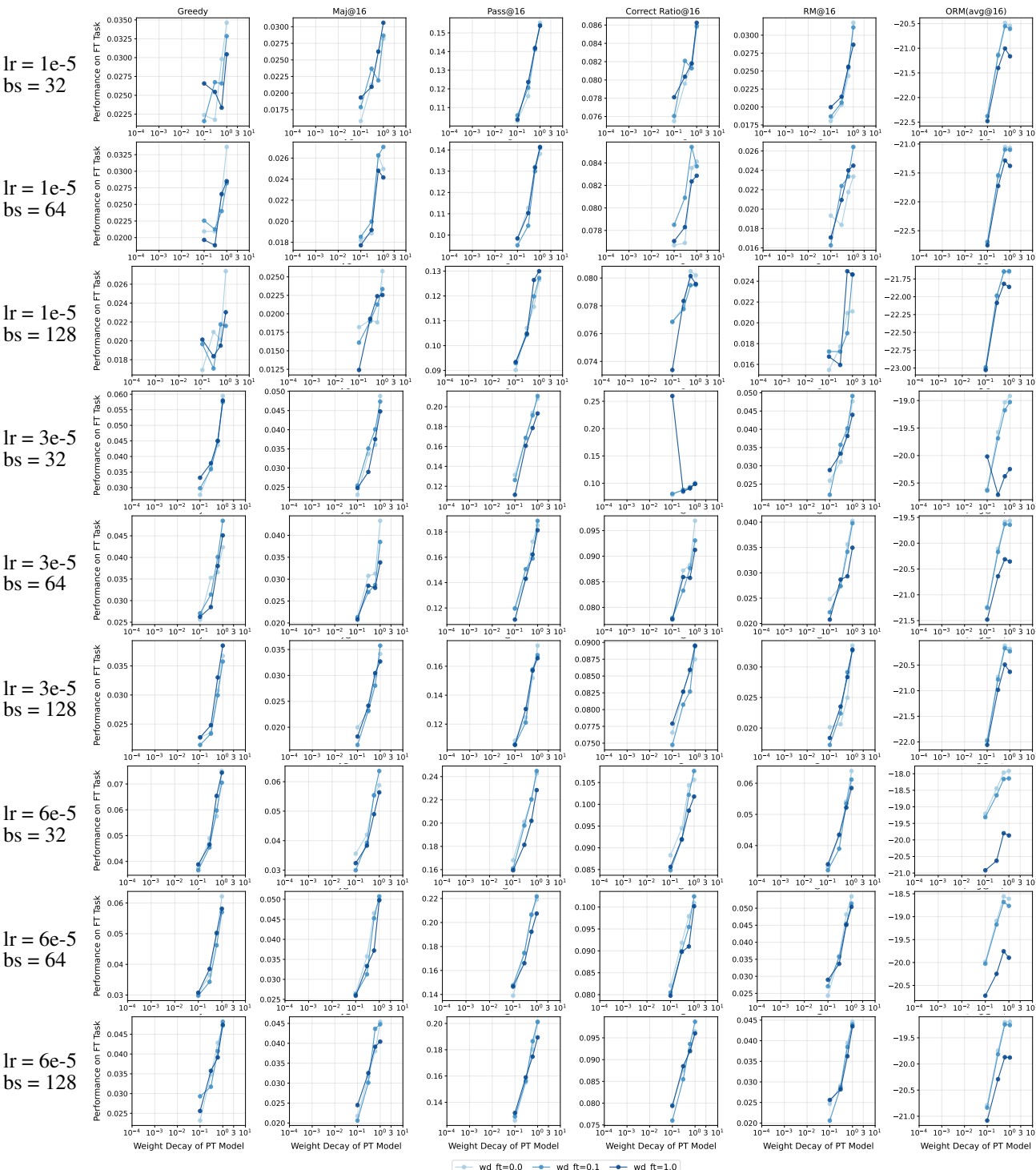

*Figure 22.* **Effect of pretraining weight decay on fine-tuning performance for each set of fine-tuning hyperparameters: SimpleScaling.** OLMo-2-1B-20x models pretrained with different weight decay values (wd_pt = [0.1, 0.3, 0.6, 1.0]) are fine-tuned on the SimpleScaling dataset. We vary the learning rate (lr = [1e-5, 3e-5, 6e-5]), weight decay (wd_ft = [0, 0.1, 1.0]), and batch size (bs = [32, 64, 128]) during fine-tuning. Average performance over all hyperparameters is shown in Figure 23. Across fine-tuning hyperparameters and evaluation metrics, the higher the weight decay during pretraining, the better the downstream performance.

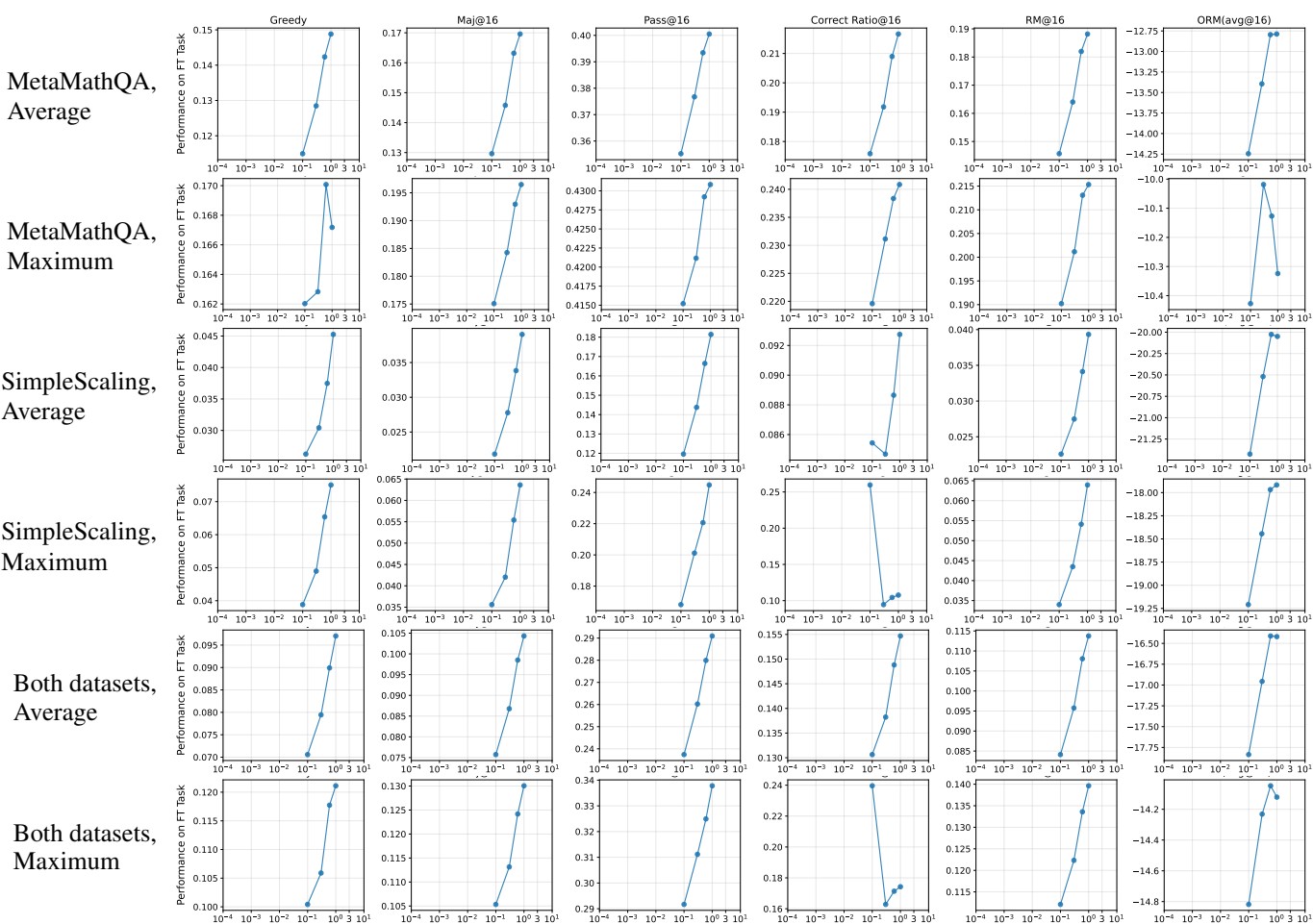

*Figure 23.* **Effect of pretraining weight decay on fine-tuning performance for the average and best set of fine-tuning hyperparameters.** This figure shows the average fine-tuning performance (Rows 1, 3, 5) and the best fine-tuning performance (Rows 2, 4, 6) over all the fine-tuning hyperparameter combinations. The higher the weight decay during pretraining, the better the fine-tuning performance.

# F. Weight Decay's Mechanistic Effects on Model Behavior

## F.1. Model representations

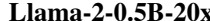

### Llama-2-0.5B-20x

### Llama-2-1B-20x

### Llama-2-4B-20x

### OLMo-2-1B-20x

### OLMo-2-1B-140x

AG News             SST             Average over datasets

*Figure 24.* **Linear probing experiments.** The left two columns, middle two columns, and right two columns show the train and test performance of the linear probes on the SST dataset, on the AG News dataset, and averaged over the two datasets, respectively.

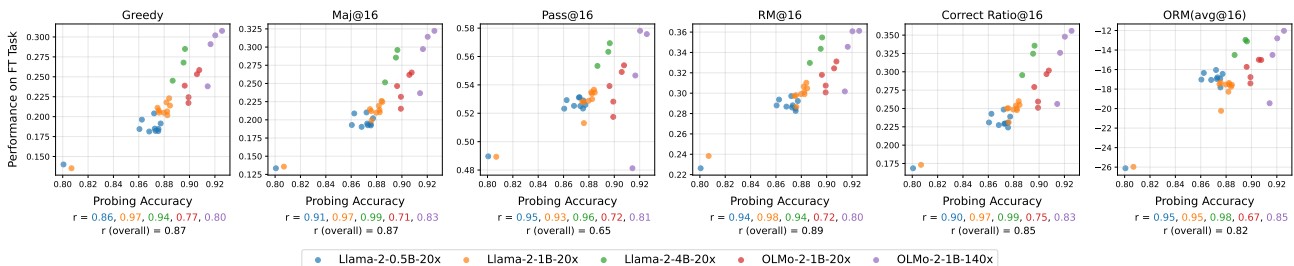

*Figure 25.* **Probing accuracy is highly predictive of downstream model performance.** The x-axis is the best average probing accuracy of the model (highest probing accuracy out of all model layers). The y-axis the average accuracy of the model over all tasks after fine-tuning. Pretrained models with higher probing accuracies from the linear probing experiments tend to perform better downstream after fine-tuning.

## F.2. Attention matrix rank

### F.2.1. ATTENTION PSEUDO-RANK COMPUTATION

To quantify the effective dimensionality of weight matrices, we follow Kobayashi et al. (2024) and compute the pseudo-rank of the matrices. For a matrix $W$ with singular values $\sigma_1 \geq \sigma_2 \geq \cdots \geq \sigma_n$, the pseudo-rank is defined as the ratio $k/n$, where $k$ is the smallest integer satisfying:

$$\frac{\sum_{i=1}^{k} \sigma_i}{\sum_{i=1}^{n} \sigma_i} \geq 0.95 \tag{4}$$

This metric represents the fraction of the largest singular values required to capture at least 95% of the sum of all singular values. In our analysis, we apply this computation to the product of the key-query matrices ($W_{QK} = W_K^T W_Q$) and the value-projection matrices ($W_{VP} = W_P W_V$) to monitor the emergence of low-rank structures during training.

### F.2.2. ADDITIONAL ANALYSES ON ATTENTION MATRIX RANK

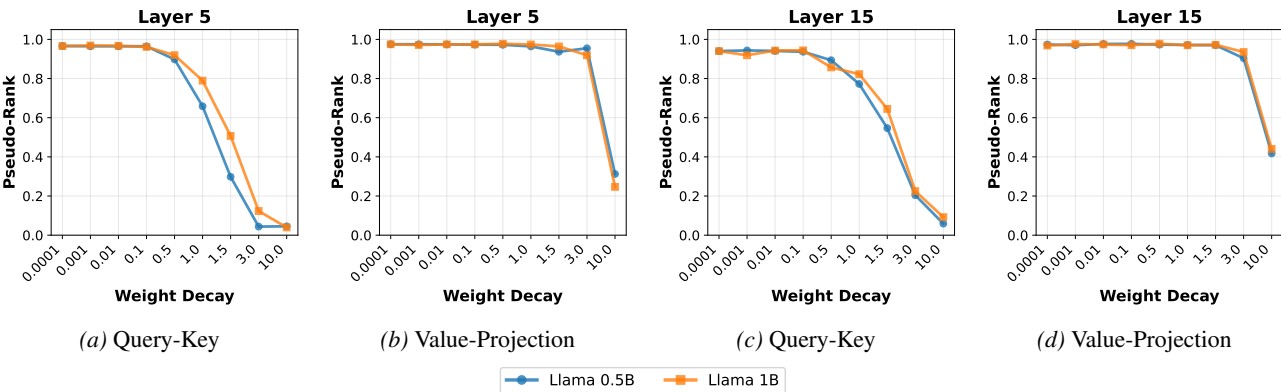

*Figure 26.* **Weight decay reduces the rank of attention matrices.** The figure depicts the average pseudo-rank (Appendix F.2.1) of the query-key ($W_{QK}$) and value projection ($W_{VP}$) matrices in layers 5 and 15 of the fully-trained Llama-2 models at 20 TPP.

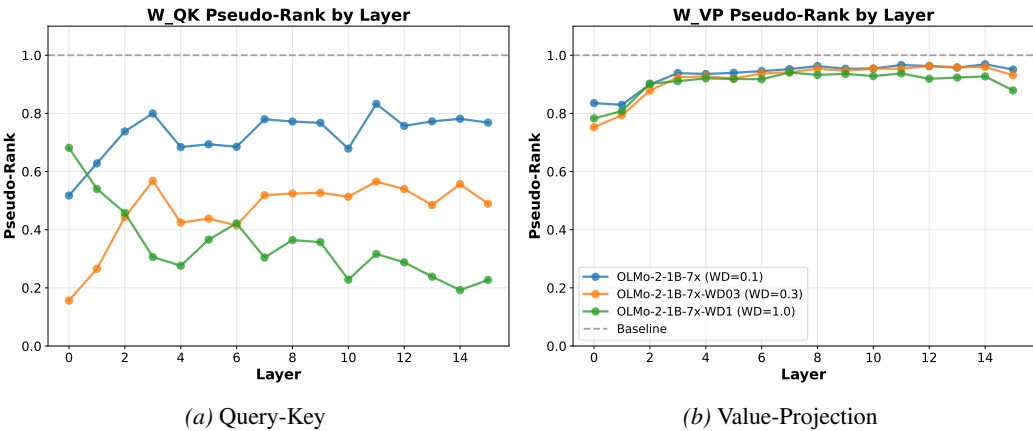

*(a)* Query-Key

*(b)* Value-Projection

*Figure 27.* **Weight decay reduces the rank of attention matrices.** This is for the OLMo models trained at 140 TPP. We observe that the rank of attention for weight decay 0.1 is generally smaller than that for both the 20 TPP and the fully trained OLMo-2-1B-0425 model. Hence, we conjecture that this is because the 140 TPP models were trained with a warmup-stable-decay learning rate schedule, whereas the 1x and 144x models were trained with a cosine learning rate schedule. While it has been shown that WSD leads to a similar validation loss to cosine decay (Hägele et al., 2024), there is emerging evidence that there are important differences between the training dynamics of the two learning rate schedules (Catalan-Tatjer et al., 2025).

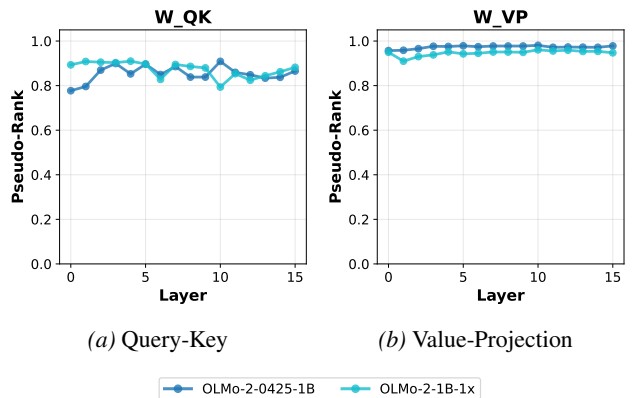

*(a)* Query-Key

*(b)* Value-Projection

*Figure 28.* **Training time does not reduce the rank of attention matrices.**

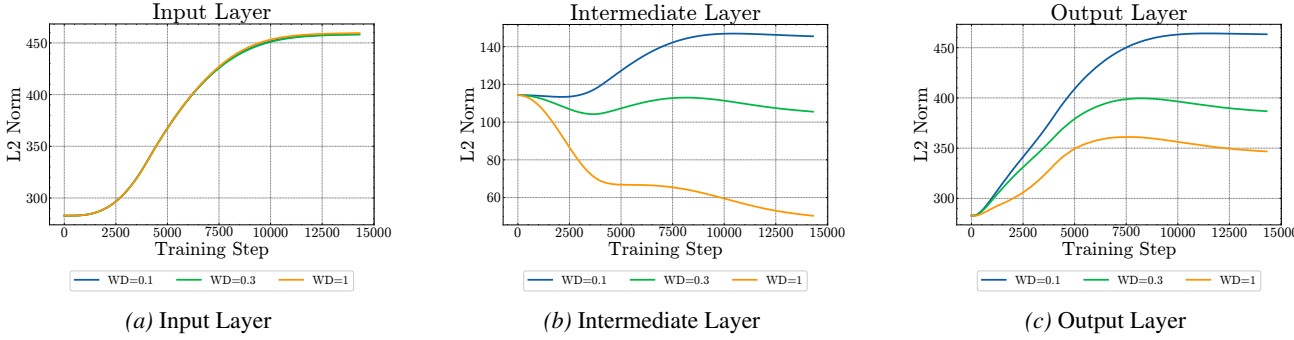

*(a)* Input Layer

*(b)* Intermediate Layer

*(c)* Output Layer

*Figure 29.* **Weight decay reduces the norm of the weights of the model.** The effect does not occur for the input layer, where the weights are not being decayed. This is for OLMo-2-1B models trained at 20 TPP.

### F.3. Norm of parameters

We also examine how weight decay changes the distribution of parameter norms across layers (Figure 30). The parameter norm of intermediate layers as a fraction of total model norm slightly decreases as weight decay increases. This is because the relative norm of lm_head (and the embedding layer which is not weight decayed for OLMo models) grows.

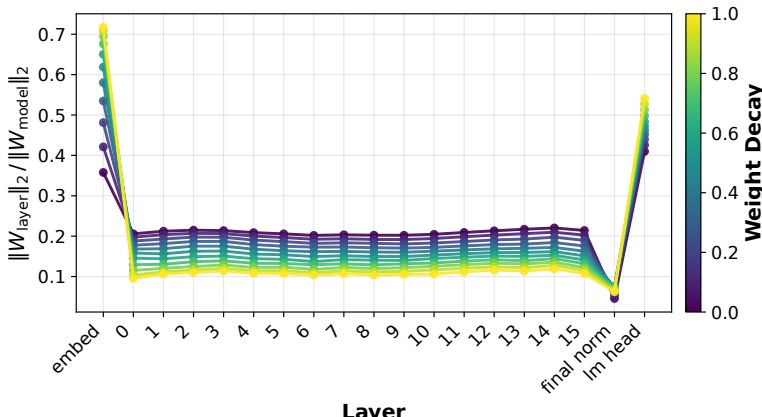

*Figure 30.* **Effect of weight decay on parameter norm.** The figure depicts the per-layer L2-norm fraction for the OLMo-2-1B-20x models. The depicted layer-wise values are the sum of all of the layers parameters.

