# OpenReview forum: "Weight Decay Improves Language Model Plasticity"
_ICML.cc/2026/Conference — ICML 2026 regular_

### Official Review · Reviewer_9MLc · 2026-02-21

**Soundness:** 2
**Presentation:** 4
**Significance:** 3
**Originality:** 3
**Overall Recommendation:** 3
**Confidence:** 2

**Summary:**

The authors strive to address the concept of model plasticity—how well a pretrained base model adapts to downstream tasks via fine-tuning—and the role of weight decay during pretraining. Through experiments on Llama-2 and OLMo-2 (up to 4B, 20 and 140 TPP), they show that larger weight decay improves plasticity: models with higher pretraining weight decay perform better after fine-tuning on six CoT tasks (six metrics). This can yield a counterintuitive regime where a base model with worse pretraining validation loss outperforms after fine-tuning. Mechanistic analysis suggests weight decay promotes linearly separable representations, regularizes attention (lower rank of W_QK/W_VP), and reduces overfitting. Overall, the research's significant contribution consists of showing that minimizing pretraining validation loss alone can fail to yield the best downstream model, and of arguing for end-to-end, plasticity-aware hyperparameter tuning.

**Compliance With Llm Reviewing Policy:**

Affirmed.

**Final Justification:**

I thank the authors for the additional experiments on OLMo and the expanded hyperparameter tuning. Although these results strengthen the manuscript, they do not fully resolve my concerns regarding the limited experimental scope and the correlational nature of the findings. The study remains restricted to specific model scales and tasks, and the mechanism by which weight decay influences rank and plasticity lacks a definitive causal or theoretical explanation. Because the underlying reasoning stays speculative and the architectural diversity is limited, I believe the paper’s generalizability is not sufficiently established. Therefore, I will maintain my Weak Reject rating.

**Key Questions For Authors:**

1. Have you varied fine-tuning strength (epochs, LR) per pretraining weight decay to see if gains hold across FT settings?
2. Do you expect similar conclusions for non-CoT downstream (e.g., classification, safety/alignment)?
3. Why might weight decay have a *particularly strong* effect on plasticity (as opposed to other pretraining hyperparameters such as learning rate or batch size)? Do you see directions for theoretical work—e.g., linking weight decay to representation geometry, optimization landscape, or stability–plasticity trade-offs—that could explain or predict this effect?

**Limitations:**

- **Scale/scope:** Llama-2 and OLMo-2 only; ≤4B params, ≤140 TPP. May not transfer to other architectures or very large/heavily overtrained models.
- **Tasks:** Six CoT tasks and six metrics; other task types (retrieval, coding, safety) and human/deployment evaluation not tested.
- **Single varying hyperparameter:** Only weight decay varied; interaction with LR, batch size, schedule not explored.
- **Mechanisms:** Linear separability, attention rank, and overfitting co-vary with weight decay and downstream performance but causality is not established (no intervention).
- **Theoretical motivation:** The paper does not provide theoretical justification for why weight decay (rather than other hyperparameters) should be especially important for plasticity; the discussion remains largely empirical.

**Strengths And Weaknesses:**

**Strengths:** Clear research question and end-to-end (pretrain → fine-tune → evaluate) design; broad experiments (two families, 0.5B–4B, two TPP regimes, six tasks/metrics) that support the main claim; important negative result that pretraining loss is not predictive of downstream performance; plausible mechanistic story (linear probing, attention rank, train–val gap) with probing–downstream correlation (Fig. 15); relevant related work.

**Weaknesses:** Scale limited to ≤4B and ≤140 TPP—unclear if the recommendation holds for much larger or longer training; sparse weight decay grid for Llama-2-4B (0.1, 1.0) and OLMo-2-1B-140x (0.1, 0.3, 1.0); correlation between pretraining loss and downstream performance is unstable (LOO); mechanisms are correlational, not causal (no intervention/ablation). Weight decay is a central pretraining hyperparameter, so its affecting plasticity is in some sense expected; the paper could better motivate *why* weight decay in particular has such a pronounced effect (relative to other hyperparameters) and whether there is or could be theoretical grounding for this relationship.

---

> ### Author Rebuttal · Authors · 2026-03-30
>
> Thank you for the insightful review and helpful feedback! We’re glad you appreciate the experiments’ “end-to-end design” and find that the research question is “clear”, the “broad experiments… support the main claim”, and the results are “important”. We address your comments and questions below.
>
> ## Q1. Vary FT hyperparameters
> This is an important point. We performed new experiments, jointly varying weight decay, learning rate, and batch size during FT. For both a fixed set and the best set of FT hyperparameters, higher weight decay leads to higher plasticity. More details in L2 of reply to Reviewer XvVs.
>
> ## Q2, L2. Settings beyond CoT
> Good point. We performed new experiments. We FT OLMo-1B-20x (wd=0.1, 0.3, 0.6, 1.0) for commonsense reasoning (5 tasks: HellaSwag, PiQA, ArcEasy, ArcChallenge, Winogrande) and safety (1 task: SafeAlpaca [1]). For both, larger weight decay tends to improve performance, consistent with CoT results. We’ll expand these experiments to other models and add them to the paper.
>
> ## Q3a, L5, W5. Why might weight decay (WD) have a stronger effect on plasticity than other pretraining hyperparameters?
> Interesting question! Indeed, prior work has shown tokenization and layer normalization also affect plasticity (Abagyan et al. 2025, Lyle et al. 2023) and it’s possible other hyperparameters do too. It’s an open question whether WD’s effect is stronger than that of other hyperparameters, but it could be possible because 1) plasticity is influenced by model weights themselves (Adams et al. 2020, Dohare et al. 2024, Kumar et al. 2023) and WD can control this more directly than other hyperparameters and 2) WD reduces the model’s intrinsic dimensionality, which has been shown to help FT [2]. This is an interesting direction for future work.
>
>
> ## Q3b, L5, W5. Potential directions for theoretical work
> Good question. We agree it’d be great to have theoretical characterization of the effects of weight decay on plasticity and model training dynamics more generally. For the rank of attention matrices, there is good theory (Kobayashi et al. 2024). However, even these analyses do not predict some of the empirical phenomena observed in our work (eg. relative difference in rank reduction between query-key and value-projection matrices). In general, a theoretical characterization of the learning dynamics of real-world language modeling has been rather difficult to achieve.
>
> For the stability-plasticity tradeoff, we performed new experiments comparing the upstream performance (7 tasks) of pretrained models and models FT’d on CoT tasks. There’s a small drop in stability (0-4 percentage points) as plasticity increases. This can motivate theoretical analyses such as characterizing the original and new learning tasks (eg. wrt task similarity and alignment with model representations) to predict change in stability.
>
>
> ## L1, W1, W2. Conclusions only hold for the examined scope
> Indeed, while our experiments (existing and new) span various model families (Llama2, OLMo2), sizes (0.5B, 1B, 1.5B, 4B), TPP ratios (20 and 140), and tasks (6 CoT, 1 safety, and 5 commonsense reasoning), results may differ beyond this scope. We will add this point to the paper.
>
> ## L3. Vary pretraining hyperparameters
> Another good point. We performed new experiments. We jointly vary weight decay (0.1, 0.6, 1.0) and learning rate (2e-4, 4e-4, 8e-4) during OLMo-1B-20x pretraining and FT models on all 6 CoT tasks (9 pretrained models, 54 setups). We find that weight decay improves plasticity across learning rates.
>
> ## L4, W4. Mechanisms are correlational
> Indeed. Establishing causality here is challenging – it’s hard to disentangle mechanisms from other covariates (eg. changing attention matrix rank while holding model representations constant). We’ll add this as a limitation. Note, while the mechanisms are correlational, the effect itself – that weight decay improves plasticity – is causal (since the experiment varies only weight decay and keeps other variables constant).
>
> ## W3. Correlation analyses
> Good point. We performed new analyses. We calculate the correlation between pretraining loss and downstream performance (six metrics, Sec3) using 35 data points (existing pretrained models and new ones from L3 above). The correlations are [-0.22, -0.25, -0.37, -0.27, -0.38, -0.51] with p-values [0.21, 0.15, 0.03, 0.11, 0.02, 0.002]. While there’s a general negative correlation, the results show that the relationship is indeed not perfect (lines 233, 298-301). Thus, optimizing only for pretraining loss or selecting a pretrained model based only on pretraining loss may lead to suboptimal final models.
>
> We will add the experiments and points discussed above to the paper. We hope our response addresses your concerns and that you consider raising the score accordingly. If you have remaining questions, we’re happy to discuss further. Thank you again!
>
> [1] Safety-tuned Llamas. ICLR 2024.
> [2] Intrinsic dim. and FT https://arxiv.org/abs/2012.13255

---

> > ### Author Rebuttal · Reviewer_9MLc · 2026-04-01
> >
> > I appreciate your extensive efforts to provide new experiments on OLMo models, additional tasks, and joint hyperparameter tuning. However, while these results strengthen the paper's main claim, I am maintaining my score of Weak Reject as the core concerns regarding the limited scope and the correlational nature of the mechanistic analysis remain fundamentally unaddressed.
> >
> > Despite the new data, the conclusions are still restricted to a specific regime of model sizes and tasks, and as you acknowledge, the work lacks a causal or theoretical bridge to explain why weight decay specifically drives these changes in rank and plasticity. While the causal link between weight decay and plasticity is clear, the underlying explanation remains speculative, and without broader architectural diversity or a more robust theoretical framework, the generalizability and depth of the contribution stay below the threshold for acceptance.

---

> > > ### Author Response · Authors · 2026-04-04
> > >
> > > Thank you for reviewing our rebuttal and for the feedback! We address the points below.
> > >
> > > **Scope:** The scope of this paper spans various model families (Llama2, OLMo2), sizes (0.5B, 1B, 1.5B, 4B), TPP ratios (20x and 140x), and downstream tasks (6 CoT, 1 safety, and 5 commonsense reasoning). We seek to explore both various setups in pretraining (we pretrain models from scratch which is particularly computationally intensive) and various model abilities during finetuning. We also jointly optimize hyperparameters when pretraining and finetuning models to examine the effect of other hyperparameters. The paper's claims apply only to the studied scope (results may differ for other scopes) and we will explicitly state this.
> > >
> > > **Correlational nature of the mechanistic analysis:** The mechanisms are indeed correlational and provide potential explanations for weight decay’s effect on plasticity. It’s hard to establish causality because it’s not possible to isolate mechanisms from other aspects of the model (eg. changing attention matrix rank while holding model representations constant). We will add this as a limitation (as previously promised). We agree that it would be great to establish causal mechanisms, but this is challenging due to the nature of deep learning models.
> > >
> > > Thank you again to the reviewer for their insights and helpful feedback for all their efforts and contributions during the review process!

---

### Official Review · Reviewer_XvVs · 2026-03-01

**Soundness:** 2
**Presentation:** 2
**Significance:** 3
**Originality:** 3
**Overall Recommendation:** 4
**Confidence:** 3

**Summary:**

This paper investigates the impact of weight decay in language model pretraining on the post-SFT performance. Authors find that the widely used default weight decay strength may not be ideal, and that the weight decay strength that leads to the best pretraining results do not always lead to the best downstream post-SFT performance. They offer three perspectives on the interaction between weight decay and training dynamics: weight decay (1) promotes a more linearly separable representation; (2) promotes low-rank representation in attention; and (3) prevents overfitting to training data.

**Compliance With Llm Reviewing Policy:**

Affirmed.

**Final Justification:**

The main strengths of this paper are the broad coverage of models (both in model family and size), and the practical implication the results have: namely, better pretraining doesn't always mean better SFT-ed models. While some of my concerns will be addressed (e.g. wd=0 model), the sample size is too small for a proper statistical analysis (e.g. correlation analysis), limiting the generalizability and robustness of the results. I increased my score to 4 after the rebuttal, which I believe reflects these points.

**Key Questions For Authors:**

1. In Figure 3, would you be able to add p-values somewhere (in the figure or in text)? I think the correlational analysis is not very suitable for sample sizes this small (e.g. $r=\pm 1$ is trivial for 2 data points).
2. Figure 4 should include at least another line which represents 0 weight decay if the main message is that weight decay, regardless of its strength, promotes linear separability w.r.t. some probing task. I saw the appendix and the picture seems more complicated, but at least wd=0 should be part of Figure 4.
3. LL349-352 right: full rank w.r.t. $d_{head}$?
4. LL369-370: At least a speculation on why $W_{QK}$ and $W_{VP}$ are affected differently would be nice.
5. Figure 6 only plots the difference - the decreasing difference can be caused by various combinations of $\Delta$train loss and $\Delta$val loss, so plotting all of them (train loss, val loss, diff) would be much more informative.

**Limitations:**

1. Small sample size could limit the generalizability of the results. For example, as pointed out in Q1 above, I honestly do not have a better idea of how to interpret the relationship between 2 points (i.e. pretraining loss and SFT result of Llama-2-4B-20x). At least this should be acknowledged in the discussion/limitation section.
2. Another thing that could potentially hurt the strength of this paper is that it may simply be the case that different weight decay during pretraining requires different hyperparameters for post-training (including SFT). If this study used the exact same SFT setup, then it's good in the sense that it's an apple to apple comparison, but not so good in the sense that we can't simply say "best pretraining loss doesn't always mean best post training results."

These limitations, together with some of the unclarities (which I asked in the question section) limit my evaluation of the paper to 3. I would be more than happy to reconsider my score if there are misunderstandings on my end; if there are sufficient clarifications!

**Strengths And Weaknesses:**

* Soundness
   * Claims seem generally supported by their experimental results. Some of the claims weren't as straightforwardly supported by others, which I outline in the question section. The small sample size is the main limiting factor, which could have been made explicit in the limitation section.

* Presentation
   * The writing is clear. I noted a few typos at the end of this section. Figures seem to be less thorough, and some counterarguments could be made about some results. I outline these points in the question section.

* Significance
   * I find the research question of this study highly impactful, striking a nice balance between being theoretically interesting and practically useful. It is true that many industry labs seem to have separate teams for pretraining and posttraining (although it's unclear how much they work with each other), and I appreciate this paper for challenging the implicit assumption that best pretraining results = best posttraining results. They also offer a few perspectives to help better understand the effect of weight decay, each of which could be an interesting venue for future research.

* Originality
   * The idea of investigating the relationship among different phases of language model training isn't new (e.g. Zhang et al., 2025; https://arxiv.org/abs/2512.07783), yet the clear and targeted focus on weight decay seems reasonable and adds original insights to the field.

* Typos
   * L155 right: predictive "of"
   * L61-62: TTP -> TPP

---

> ### Author Rebuttal · Authors · 2026-03-30
>
> Thank you for the insightful review and helpful feedback! We’re glad you find that the research question is “highly impactful, striking a nice balance between being theoretically interesting and practically useful” and that the paper “challeng[es] the implicit assumption” in language model training, “offer[s] a few perspectives to help better understand the effect of weight decay”, and “adds original insights to the field”. We address your comments and questions below.
>
> ## Q1. New correlation analyses
> This is a good point. We agree that the existing correlation analyses (which sometimes use a few points) are not ideal. To measure the relationship between pretraining loss and downstream performance, we performed new analyses, calculating the correlation between pretraining loss and each measure of downstream performance (six metrics in Sec3). Each correlation is calculated using 35 data points, i.e., 35 pretrained models (from existing experiments and new ExpA in the response for Reviewer FspF). The correlations are [-0.22, -0.25, -0.37, -0.27, -0.38, -0.51] with p-values [0.21, 0.15, 0.03, 0.11, 0.02, 0.002]. While there is a general negative correlation, the results support the point that this relationship is not perfect (lines 233 and 298-301). Thus, optimizing for pretraining loss alone or selecting one pretrained model over another based only on pretraining loss (as are common in practice) may lead to suboptimal final models. We will use these new analyses instead. Thank you for this helpful idea!
>
>
> ## Q2. Training new wd=0 models
> Another good point. We will pretrain a wd=0 model for each model family, perform downstream experiments for these models, and add them to the final paper.
>
>
> ## Q3. Full rank w.r.t. $d_{head}$?
> Yes.
>
> ## Q4. Why might weight decay affect $W_{QK}$ and $W_{VP}$ differently?
> Great question. While it is speculation, low-rank $W_{QK}$ could mean that individual attention heads become specialized, which may be beneficial for learning and generalization. In contrast, low-rank $W_{VP}$ restricts the flow of information from attention heads back to the residual stream, which may be less desirable. This is an interesting point of discussion and we will add it to the paper.
>
> ## Q5. Additional analyses for train loss, validation loss, and difference (Figure 6)
> Another great question. The validation loss is depicted in Figure 1b, and the train loss follows a similarly U-shaped curve that lies below the validation loss. We will add all curves to the paper.
>
> ## L1. Small sample size
> Indeed, we pretrain fewer models for some model families (Llama-2-4B and OLMo-1B-140x) due to the high amount of compute required for their pretraining. However, for these models, the results are consistent with other more extensively studied models: 1) better pretraining validation loss does not guarantee better downstream performance and 2) the weight decay that leads to best downstream performance is larger than the standard 0.1 default. Nonetheless, we will add this limitation to the paper.
>
> ## L2. Vary FT hyperparameters
> This is an important point. We performed new experiments. For pretrained OLMo-1B-20x models (wd_pt=[0.1, 0.3, 0.6, 1.0]), we vary weight decay (0, 0.1, 1.0; wd_ft), learning rate (1e-5, 3e-5, 6e-5; lr_ft), and batch size (32, 64, 128) during finetuning for two tasks (MetaMathQA, SimpleScaling), leading to 216 model setups. wd_ft=0.6 and lr_ft=6e-5 setups are still running, but other setups are complete. We examine the effect of pretraining weight decay for a fixed set of FT hyperparameters and find that higher pretraining weight decay leads to better FT performance. We also examine the effect of pretraining weight decay for the best set of FT hyperparameters (i.e., the ones that lead to the best FT performance for a given pretrained model; what one would be most interested in in practice) and find that higher pretraining weight decay also leads to better FT performance.
>
> We will add the experiments and points discussed above to the paper. We hope our response addresses your concerns and that you consider raising the score accordingly. If you have remaining questions, we are happy to discuss further. Thank you again!

---

> > ### Author Rebuttal · Reviewer_XvVs · 2026-04-03
> >
> > Thank you for the thorough response! Most of my conerns are fully resolved, and I'm adding a few follow-up questions.
> > * 1: If I'm not mistaken, the correlation analysis only makes within each model size, correct? I think by design, increasing the number of models per scale is the only way out - (1) currently the number of models for each size family is too small, leading to trivial correlation analyses (e.g. $r=-1$ with 2 data points), and (2) because model size affects the loss, an aggregate correlation across different model sizes would be noisy.
> > * 2: Noted, thank you. If the results are ready, I would love to know what they look like!
> > * 3, 4, 5, L1: Noted, thank you.
> > * L2: This is very helpful. I think highlighting this in the final revision would make the paper much stronger!

---

> > > ### Author Response · Authors · 2026-04-04
> > >
> > > Thank you for the continued feedback! We address the points below.
> > >
> > > **Q2. Pretraining wd=0 models.** The new wd=0 models have not finished pretraining (the jobs are queuing and queuing for longer than before due to our earlier heavy use of the cluster to pretrain, finetune, and evaluate models for rebuttal experiments). The models take up to 2 weeks to pretrain, after which they will also be further finetuned and evaluated on downstream tasks. For some models that have pretrained variants with weight decay near zero (eg. wd={1e-4, 1e-3, 1e-2}), we hypothesize that the performance of the wd=0 pretrained model will be similar. For all models, we believe these wd=0 additions will provide insightful information and will be sure to add the results to the final version. Thank you again for this helpful idea!
> > >
> > > **Q1. Correlation analyses.** This is a good point. We will carefully evaluate how to discuss the correlations. One option is to provide per-group correlations (each group will now have more data points due to new pretrained models from the experiment above and from ExpA in response to Reviewer FspF) and qualify the interpretation by noting the small samples of some groups. The *key point* is that better pretraining loss “is not perfectly predictive of” and “does not necessarily lead to” better downstream performance (eg. line 233 and the Takeaway box in lines 297-302). We can see this point even without correlation values: for most model groups (ie. combinations of model family, size, and TPP), we see examples of a model with worse pretraining loss having better downstream performance than a model with better pretraining loss (we also see examples of models with similar pretraining loss having different downstream performance). This key point is important because it may be counterintuitive in the current language modeling paradigm and can inform the way we optimize and choose among models (since it suggests that optimizing for pretraining loss alone or selecting one pretrained model over another based only on pretraining loss, as sometimes done in practice, may lead to suboptimal final models). We will further discuss this key point in the paper, incorporating the feedback from our two rounds of discussion.
> > >
> > > Thank you again to the reviewer for their key insights and invaluable feedback which have meaningfully improved the paper and for all their efforts and contributions since the very beginning of the review process!

---

### Official Review · Reviewer_FspF · 2026-03-11

**Soundness:** 3
**Presentation:** 3
**Significance:** 2
**Originality:** 3
**Overall Recommendation:** 5
**Confidence:** 4

**Summary:**

This paper studies how the weight decay hyperparameter during language model pretraining affects model plasticity, defined as the ability of a pretrained model to adapt to downstream tasks through fine-tuning. The authors systematically vary weight decay during pretraining across several model families (Llama-2 and OLMo-2), model sizes (up to 4B parameters), and training regimes. They then finetune the resulting models on six reasoning tasks. The main finding is that larger weight decay during pretraining often leads to better downstream performance after finetuning, even when it slightly worsens pretraining validation loss. The paper also provides mechanistic analyses suggesting that weight decay encourages more linearly separable representations, reduces attention matrix rank, and mitigates overfitting during pretraining.

**Compliance With Llm Reviewing Policy:**

Affirmed.

**Final Justification:**

My main concerns are: (1) the scope of the downstream tasks selected by the authors may differ substantially from the knowledge learned during pretraining, and (2) the observed downstream task performance is mostly negatively correlated with the pretraining loss.
The authors have addressed the first issue through additional experiments. For the second issue, they argue that pretraining loss cannot perfectly predict downstream task performance. Although I still find this conclusion somewhat weak and not entirely satisfying, what the authors actually claim is not about arbitrary pretraining loss, but rather that an increase in pretraining loss caused by weight decay does not necessarily imply a decrease in downstream task performance.

In my view, this is consistent with basic statistical intuition and is a point that has been overlooked in the era of large models. Therefore, I have raised my score to 5.

**Key Questions For Authors:**

- How sensitive are the results to other optimizer hyperparameters such as learning rate schedules or batch size?
- For tasks that are more related to the pertaining data, is the conclusion still true?
- In Figure 4, the improvement from larger weight decay seems to be concentrated mainly in the lower layers, while the effect on higher-layer linear probing accuracy appears much smaller. Could the authors analyze how weight decay changes the distribution of parameter norms across layers? For example, it would be useful to report the relative weight norms (or weight norm ratios across layers) under different weight decay values, to better understand whether stronger weight decay disproportionately affects lower-layer representations.

**Limitations:**

Yes

**Strengths And Weaknesses:**

***Strength***
- The paper highlights an important but underexplored issue: selecting pretraining hyperparameters based solely on validation loss may not optimize downstream adaptability.
- The experiments cover multiple model families, scales, and training regimes, providing relatively broad empirical evidence.
- The work includes several mechanistic analyses (linear probing, attention rank, overfitting metrics), which help provide intuition for the observed effects.
- The results have potential practical implications for how pretraining hyperparameters are chosen in modern LLM pipelines.

---

***Weakness***
- Downstream evaluation focuses mainly on Chain-of-Thought reasoning tasks, limiting the diversity of tasks considered. It is unclear whether the same trends would hold for tasks that are more related to the pretraining data.
- The mechanistic explanations are largely correlational and do not establish clear causal mechanisms linking weight decay to plasticity.
- The claim that pretraining loss is not predictive of downstream performance may be somewhat overstated. In Figure 3, except for the OLMo-2-1B-140x setting (which only contains three data points), most configurations still show negative correlations between pretraining loss and downstream accuracy, suggesting that pretraining loss remains a reasonably informative proxy in many regimes.

---

> ### Author Rebuttal · Authors · 2026-03-30
>
> Thank you for the insightful review and helpful feedback! We’re glad you find that the research question is “important but underexplored” and that the findings are substantiated by “broad empirical evidence” and “help provide intuition” and have “practical implications” for pretraining. We address your comments and questions below.
>
> ## W1 & Q2. Explore tasks more related to pretraining
> This is a good point. We performed new experiments examining 1) commonsense reasoning (5 tasks: HellaSwag, PiQA, ArcEasy, ArcChallenge, Winogrande) which is more related to pretraining since tasks can be solved based on patterns and knowledge from pretraining without explicit reasoning, and 2) safety (1 task: SafeAlpaca [1]). We finetune OLMo-1B-20x models (wd=0.1, 0.3, 0.6, 1.0) for these tasks and evaluate performance. For both new model abilities, larger weight decay tends to yield better performance, consistent with the results for CoT. We will expand these experiments to all other models and add them to the paper.
>
> ## W2. Mechanistic explanations are correlational
> Indeed, these explanations are correlational. Establishing causality here is challenging – it is hard to disentangle mechanisms from other covariates (e.g., changing attention matrix rank while holding model representations constant). We will add this as a limitation. Note that, while the mechanisms are correlational, the effect itself – that weight decay improves plasticity – is causal (since the experiment varies only weight decay and keeps other variables constant).
>
> ## W3. Relationship between pretraining loss and downstream performance
> We agree there is a general relationship between pretraining loss and downstream performance. We performed new analyses, calculating the correlation between pretraining loss and each measure of downstream performance (six metrics in Sec3) for 35 pretrained models (from existing experiments and ExpA below). The correlations are [-0.22, -0.25, -0.37, -0.27, -0.38, -0.51] with p-values [0.21, 0.15, 0.03, 0.11, 0.02, 0.002]. Thus, to your point, there is indeed a negative correlation between the two.
>
> What we aim to highlight in the paper is that this relationship is not perfect/monotonic, i.e., better pretraining loss “is not perfectly predictive” and “does not necessarily lead to” better downstream performance (lines 233 and 298-301). Thus, optimizing for pretraining loss alone or selecting one pretrained model over another based only on pretraining loss (as are common in practice) may lead to suboptimal final models. We will clarify this.
>
> ## Q1. Examine other training hyperparameters
> Another good point. We performed new experiments investigating the role of pretraining (ExpA) and FT (ExpB) hyperparameters.
>
> *ExpA.* We vary weight decay (0.1, 0.6, 1.0) and learning rate (2e-4, 4e-4, 8e-4) during OLMo-1B-20x pretraining and FT models on all six CoT tasks, yielding 9 pretrained models and 54 setups. We find that, across learning rates, higher weight decay yields better FT performance, consistent with results in the paper.
>
> *ExpB.* For pretrained OLMo-1B-20x models (wd_pt=[0.1, 0.3, 0.6, 1.0] with default learning rate 4e-4), we vary weight decay (0, 0.1, 1.0; wd_ft), learning rate (1e-5, 3e-5, 6e-5; lr_ft), and batch size (32, 64, 128) during FT for two tasks (MetaMathQA, SimpleScaling), yielding 216 model setups. wd_ft=0.6 and lr_ft=6e-5 setups are still running, but other setups are complete. We examine the effect of pretraining weight decay for a fixed set of FT hyperparameters and find that higher pretraining weight decay leads to better FT performance. We also examine the effect of pretraining weight decay for the best set of FT hyperparameters (i.e., the ones that lead to the best FT performance for a given pretrained model; what one would be most interested in in practice) and find that higher pretraining weight decay also leads to better FT performance.
>
>
> ## Q3. How does weight decay change the distribution of parameter norms across layers?
> Interesting question. We performed new analyses. The weight norm of intermediate layers as a fraction of total model norm slightly decreases as we increase the weight decay value (approximately 0.08 -> 0.06). This is because the relative norm of lm_head (and in case of OLMo models where it is not decayed, the embedding layer) grows. However, we find no discernable trend in lower layers.
>
>
> We will add the experiments and points discussed above to the paper. We hope our response addresses your concerns and that you consider raising the score accordingly. If you have remaining questions, we are happy to discuss further. Thank you again!
>
> [1] Safety-tuned llamas. ICLR 2024.

---

> > ### Author Rebuttal · Reviewer_FspF · 2026-04-02
> >
> > My concerns have been adequately addressed.  I will increase my score to 5.

---

> > > ### Author Response · Authors · 2026-04-04
> > >
> > > We sincerely appreciate your raising the score after reviewing our rebuttal. This paper greatly benefited from your feedback and we will incorporate it in the final version. Thank you for all your efforts and contribution throughout the review process!

---

### Official Review · Reviewer_iBxP · 2026-03-11

**Soundness:** 3
**Presentation:** 3
**Significance:** 4
**Originality:** 3
**Overall Recommendation:** 4
**Confidence:** 4

**Summary:**

This paper studies a meaningful and underexplored question in LLM training: whether pretraining hyperparameters should be chosen for downstream adaptability rather than solely for pretraining validation loss. It focuses on weight decay and shows, across Llama-2 and OLMo-2 models, multiple scales, and both 20-TPP and 140-TPP regimes, that large pretraining weight decay can improve downstream fine-tuning performance even when it does not minimize pretraining loss. The paper further argues that this effect is associated with more linearly separable representations, reduced attention rank, and less overfitting.

**Compliance With Llm Reviewing Policy:**

Affirmed.

**Final Justification:**

I recommend a weak accept for this paper for the following reasons.

Goodside:
- The paper addresses a worthwhile question: in current practice, pretraining hyperparameters are typically optimized for pretraining performance, while downstream performance is often overlooked, which authors name as model plasticity.
- The empirical study is broad and covers a range of model scales and experimental settings.

I do not raise my score further for three reasons:
- The main empirical observation—that larger weight decay can improve downstream performance despite hurting pretraining performance—is already broadly recognized. Although prior work has not focused specifically on theLLM  pretraining setting considered here, the work pushes further to LLM pretraining
- The theoretical perspective is not entirely novel, though it does extend and sharpen existing understanding.
- The work provide a good but not strong enough actionable guideline for practical experiments.

**Key Questions For Authors:**

1. Can the authors provide a more actionable rule for choosing weight decay?
2. How robust are the conclusions beyond SFT on CoT-style tasks, e.g., for broader instruction tuning or RL-based post-training?
3. Which parts of the mechanistic account are genuinely new in terms of thoery, beyond previously known effects of weight decay on generalization and attention rank?
4. Can the authors improve the main figures and adopt more standard notation VO?

**Limitations:**

yes

**Strengths And Weaknesses:**

**Strengths**

* The paper asks a **worthwhile** question: current practice largely optimizes pretraining hyperparameters for base model cross-entropy loss, whereas this work evaluates them through the lens of downstream plasticity. This is a valuable perspective shift.
* The empirical coverage is fairly **broad**: spanning two model families, multiple scales up to 4B, two training regimes.

**Weaknesses**

While the paper reports several interesting findings, I remain on the borderline between accept and reject for the following reasons.

* Despite the interesting findings, the paper offers limited practical guidance for tuning weight decay in real LLM training and new theory about weight decay. The study shows that larger values can help, but does not yield a clear **actionable rule** or provide a **novel** theory.
* `Weight decay reduces the rank of attention matrices` and `Weight decay reduces overfitting on training data` is **well-known**. The present paper merely extends these observations to a larger scale setting.
- In the LLM setting, existing work has also suggested that larger weight decay can improve compositional generalization [1], very similar to your findings.
* Presentation needs improvement.
   * igures 1(a) and 2 are very plausible. In particular, it is unclear how to read the regimes between $(0.1, 1)$ and $(1, 10)$. Moreover, some curves in Figure 1(c) and Figure 2 contain too few points to support a convincing trend.
   * The notation $W_{VP}$ is very unusual, $W_{VO}$ would be more standard [2,3].

[1] Complexity Control Facilitates Reasoning-Based Compositional Generalization in Transformers. TPAMI. \
[2] The Sharpness Disparity Principle in Transformers for Accelerating Language Model Pre-Training. ICML 2025. \
[3] Muon Outperforms Adam in Tail-End Associative Memory Learning. ICLR 2026.

---

> ### Author Rebuttal · Authors · 2026-03-30
>
> Thank you for the insightful review and helpful feedback! We appreciate your recognition of the “worthwhile” research question, the paper’s “valuable perspective shift”, and the “broad” span of experiments. We address your comments and questions below.
>
> ## W1 & Q1. Rule to choose weight decay and theory
> We agree that an actionable rule for choosing the weight decay value would be great. Indeed, previous work has suggested that the weight decay value may be chosen by fitting a scaling law of the weight decay value against the pretraining loss (Bergsma et al. 2025). Our results complicate this story since we show that pretraining loss may not always be the best target for optimizing the weight decay value if one cares about downstream performance. Of course, future work may now attempt to derive scaling laws for weight decay values against downstream performance. However, obtaining such scaling laws has generally been harder than for the pretraining loss.
>
> We agree that it would be great to have a theoretical analysis to complement our empirical results. However, since it has generally been very hard to obtain theoretical results that reliably describe the training dynamics of real-world language modeling, we don’t see the absence of theory as a significant drawback of the paper. Nonetheless, we'll add this discussion to the paper. To strengthen empirical results, we performed new experiments investigating model abilities beyond CoT, the role of pretraining hyperparameters, and the role of FT hyperparameters (see Q2 below, and ExpA and ExpB in the response to Reviewer FspF). The results from these new experiments provide further evidence supporting the paper’s claims.
>
>
> ## W2 & Q3. Weight decay’s effects on attention matrix rank and overfitting
> We agree that weight decay’s effects on attention matrix rank and overfitting have been observed in previous work. However, we would like to highlight that our work not only replicates previous results in a larger setting, but also uncovers novel phenomena that have not been previously identified. Unlike what is suggested by Kobayashi et al. (2024), we find that a reduction in the rank of attention matrices is correlated with better pretraining and fine-tuning performance (up to a weight decay of 0.6 for the OLMo models). In addition, we find that the Query-Key and Value-Projection matrices are differentially affected by weight decay, which also has not been reported before.
>
>
> ## W3. Comparison with [1]
> Indeed [1] and our work have connections but the two works also differ in fundamental ways. Both works study weight decay’s role in model behavior, but [1] studies its role in OOD compositional generalization and reasoning-vs-memorization transitions under complexity control, while our work focuses on its role in language model plasticity in the realistic pretrain-then-finetune setting and shows that higher weight decay can improve downstream performance even if it worsens pretraining loss. Nonetheless, these two works are conceptually compatible and we will add this paper to the related work section.
>
> ## W4 & Q4. Figures and notation
> Yes, we will improve figures and use the suggested notation. In Fig.1c and 2, there are fewer Llama-4B and OLMo-1B-140TPP models (and thus fewer data points) due to the large compute required for their pretraining. However, for these models, the results are consistent with other more extensively studied models: 1) better pretraining loss does not guarantee better downstream performance and 2) the weight decay that leads to best downstream performance is larger than the standard 0.1 default.
>
> We will perform new experiments to examine an even wider weight decay range (this will also add more data points). We will pretrain a wd=0 model for each model family, perform downstream experiments on these models, and add results to the final paper.
>
> ## Q2. Experiments beyond CoT
> This is a good point. We performed new experiments, finetuning OLMo-1B-20x (wd=0.1, 0.3, 0.6, 1.0) for commonsense reasoning (5 tasks: HellaSwag, PiQA, ArcEasy, ArcChallenge) and safety (1 task: SafeAlpaca [2]). For both model abilities, we observe that larger weight decay tends to yield better performance, consistent with CoT results. We will expand these experiments to all other models and add these results to the paper.
>
> We will add the experiments and points discussed above to the paper. We hope our response addresses your concerns and that you consider raising the score accordingly. If you have remaining questions, we are happy to discuss further. Thank you again!
>
>
>
> [1] Complexity Control Facilitates Reasoning-Based Compositional Generalization in Transformers. TPAMI.
>
> [2] Safety-tuned Llamas. ICLR 2024.

---

> > ### Author Rebuttal · Reviewer_iBxP · 2026-04-01
> >
> > I’ve read the rebuttal. I was initially on the borderline between accept and reject, and I now lean toward weak accept. \
> > I did not raise my score further for the following reasons:
> > - [1, 2] observe that increasing the head weight decay during pre-training can hurt **upstream (US) performance** while improving **downstream (DS) performance**. (not mentioned by paper)
> > - [3] notes that the weight decay used in self-supervised learning significantly affects **transferability**, and that selecting an appropriate value during upstream pre-training can be challenging; downstream evaluation is therefore important. (mentioned by paper)
> >
> > Although these works are not in exactly the same setting, they share a **similar** underlying intuition with your paper.
> >
> > Besides, the actionable guidance provided by the rebuttal is good but not strong enough.
> >
> > I recommend that the authors include a more explicit discussion of all these things.
> >
> > ---
> > **Ref**
> >
> > [1] Scaling vision transformers. \
> > [2] Exploring the limits of large scale pre-training. \
> > [3] Rethinking Evaluation Protocols of Visual Representations Learned via Self-supervised Learning.
> >
> > ---
> > # Reply to authors' reply below
> > # I find a new reply is not visible to author, so I add it here
> >
> > ---
> > Thank you for the clarification.
> >
> > However, in [1, 2], the **upstream task** actually refers to the **pretraining task**, so **upstream performance** should correspond to **pretraining performance (loss)**. Besides, in my understanding, [1, 2] mean that weight decay during pretraining **can** hurt upstream (US) performance while improving downstream (DS) performance, rather than implying that this always happens.
> >
> > Hence, I think your reply has misunderstanding? [1, 2] should also share a similar underlying intuition with your paper.

---

> > > ### Author Response · Authors · 2026-04-04
> > >
> > > Thank you for reviewing our rebuttal and for raising new points to help improve the paper! We address them below.
> > >
> > > **Can weight decay hurt upstream performance while improving downstream performance?** We performed additional experiments and did not observe this phenomenon. We evaluate the upstream performance of the pretrained models (5 tasks: PiQA, Winogrande, HellaSwag, ArcEasy, ArcChallenge; we report average upstream accuracy) and observe the following:
> > >
> > > Llama-0.5B-20x, wd = {0.1, 0.5, 1.0} -> upstream accuracy = {0.40, 0.41, 0.41*}
> > >
> > > Llama-1B-20x, wd = {0.1, 0.5, 1.0} -> upstream accuracy = {0.44, 0.45, 0.45*}
> > >
> > > OLMo-1B-20x, wd = {0.1, 0.3, 1.0} -> upstream accuracy = {0.49, 0.49, 0.50*}
> > >
> > > OLMo-1B-140x, wd = {0.1, 0.3*, 1.0} -> upstream accuracy = {0.61, 0.60*, 0.56}
> > >
> > > *weight decay value of pretrained model with best downstream performance
> > >
> > > Thus, in contrast to [1,2] which study image models, we find that the higher downstream performance offered by weight decay may not always be accompanied by lower upstream performance for language models (in the results above, upstream performance was not affected). We will add [1,2] and these experiments to the paper.
> > >
> > >
> > >
> > > **Paper [3]:** Indeed, [3] finds that weight decay in self-supervised learning (SSL) improves the transferability of image model representations and that it is challenging to find the ideal weight decay value because this benefit of weight decay is not captured by existing SSL evaluation methods. While the setting of our paper is different (we study the role of weight decay in model plasticity for language models), the results in [3] are compatible with ours and we will add this discussion to the paper.
> > >
> > > We will add the points from both rounds of discussion to the paper. Thank you again to the reviewer for your insights and helpful feedback throughout the review process!
> > >
> > > [1] Scaling vision transformers.
> > >
> > > [2] Exploring the limits of large scale pre-training.
> > >
> > > [3] Rethinking Evaluation Protocols of Visual Representations Learned via Self-supervised Learning.

---

### Decision · Program_Chairs · 2026-04-30

**Decision:**

Accept (regular)

**Comment:**

The paper studies how pretraining weight decay affects a model's ability to adapt during fine-tuning, and shows across two model families and several scales that larger weight decay can improve downstream performance even when it slightly worsens pretraining loss. Reviewers agreed the question is worthwhile and the end-to-end study is broad enough to support the main claim. The principal reservation was originality: similar upstream-versus-downstream tradeoffs have been observed in the vision literature, and the mechanistic analyses are correlational rather than explanatory.

The rebuttal does not produce a causal or theoretical account, and the authors are right that one is hard to obtain here, but it does show the effect is robust to the obvious confounds, holding across additional task families and across pretraining and fine-tuning hyperparameter choices. After discussion three of four reviewers support acceptance, and the remaining concern is about depth of explanation rather than the validity of the finding. I recommend acceptance.

For camera-ready, please include the experiments promised in the rebuttal (including the wd=0 models), add the related vision-scaling work to the discussion, and sharpen the headline claim along the lines the reviewer FspF suggested.

Congratulations!